# Transient Anticyclonic Eddies and Their Relationship to Atmospheric Block Persistence

Charlie C. Suitters[1], Oscar Martínez-Alvarado[1,2], Kevin I. Hodges[1,2], Reinhard K. H. Schiemann[1,2], and Duncan Ackerley[3]

[1]Department of Meteorology, University of Reading, Reading, United Kingdom
[2]National Centre for Atmospheric Science (NCAS), University of Reading, Reading, United Kingdom
[3]Met Office, Exeter, United Kingdom

**Correspondence:** Charlie Suitters (c.c.suitters@pgr.reading.ac.uk)

**Abstract.** Atmospheric blocking is a circulation pattern that describes the presence of large-scale, persistent anticyclones, which have the potential to bring severe impacts at the surface. However, the dynamical behaviour of blocks is still not fully understood. For example, the factors that determine the persistence of blocking events are not clear. In this study, the relationship between blocks and smaller-scale transient anticyclonic eddies is examined, with a particular focus on the impact of transients on the persistence of a block. Analysis is performed in two areas: the Euro-Atlantic and North Pacific, which are locations with both high blocking frequency and potential for severe impacts. Geopotential height anomalies at 500 hPa are used to identify blocking events and the anticyclonic transient eddies. This allows for a Eulerian definition of blocking, as well as a Lagrangian perspective on the eddies. It is found that anticyclonic eddies experience a northward acceleration prior to entering a block, which is indicative of ridge-building ahead of the block, but could also potentially provide evidence for the previously-proposed Selective Absorption Mechanism for block maintenance. A general pattern is found whereby longer blocks interact with more anticyclonic transients than less persistent blocks at all times of year. This effect is strongest in winter and weakest in summer, which agrees with the fact that blocks are most persistent in winter and least persistent in summer. However, the strength of the anticyclonic eddy that interacts with a block, measured by its maximum 500 hPa geopotential height anomaly, has a more complicated relationship with block persistence. The strength of anticyclonic transient eddies is a more determining factor of block persistence in the North Pacific than in the Euro-Atlantic region. In the North Pacific the longest blocks interact with stronger eddies than the shortest blocks in all seasons except summer, when the reverse is true. By contrast, longer Euro-Atlantic blocks only result from stronger anticyclonic eddies in autumn and winter. We therefore conclude that the number of anticyclonic eddies that interact with a block is most important in determining its persistence, with the strength of the eddies having a more variable effect.

## 1 Introduction

Extratropical atmospheric blocking is important due to the anomalous, and sometimes severe, weather conditions that are often observed at the surface. These conditions occur due to anomalously large and slow-moving high pressure systems characteristic

of blocking weather patterns, which act to disrupt the climatological mid-latitude zonal flow, instead diverting it to the north and south (Rex, 1950).

A complete dynamical understanding of block formation, maintenance, and decay is still lacking. One of the main reasons for this is the difficulty in defining what constitutes a block (Woollings et al., 2018) due to the disparate synoptic conditions that can be described as "blocked". Compounding this issue further is the fact that there are numerous ways to define and detect a block in gridded datasets (Barriopedro et al., 2010; Woollings et al., 2018), and no one method detects every weather pattern that a synoptician would identify as a block. However, the overwhelming consensus is that blocking events are qualitatively

described by their high surface pressure, persistence, large spatial area, and quasi-stationarity (Rex, 1950), with blocks typically lasting 1-2 weeks (e.g., Woollings et al., 2018; Lupo, 2021; Kautz et al., 2022).

With these characteristics in mind, previous studies have been able to identify a range of mechanisms that are important for blocking, with most of the focus on block formation processes. Large-scale wave dynamics are a principal way in which blocks can form, for example through a simple stationary ridge in the planetary wave pattern (Legras and Ghil, 1985), constructive

interference of waves with different scales (Austin, 1980; Shutts, 1983), or Rossby wave breaking (e.g., Altenhoff et al., 2008; Masato et al., 2012). Other processes important to block formation include rapid cyclogenesis (Colucci, 1985; Nakamura and Huang, 2018) and diabatic heating (Pfahl et al., 2015; Lenggenhager and Martius, 2020; Zschenderlein et al., 2020). The relative importance of these mechanisms varies by location (Miller and Wang, 2022), and can even vary within a region (Drouard and Woollings, 2018), which suggests that block formation is complex. More recently, there have been studies on

the specific processes for block maintenance, in an attempt to explain why some blocks persist for longer than others. For example, Drouard et al. (2021) examined Northern Hemisphere (NH) blocks and concluded that the most important factor in determining block persistence was the direction of the Rossby wave breaking. Cyclonically-breaking blocks tend to be longer lived than blocks that form from anticyclonic wave breaking. However, their study assumes that all blocks can be classified by the direction and morphology of wave breaking, when in fact a blocking event can take many different shapes (i.e., ridge,

omega, dipole/Rex, cyclonic and anticyclonic wave breaking) during its lifetime (e.g., Woollings et al., 2018; Sousa et al., 2021).

It has long been known that there is a two-way interaction between blocking systems and smaller-scale synoptic transient eddies. Blocks cause synoptic eddies to slow down and stall, and these same eddies can enhance and maintain the block (e.g., Shutts, 1983; Mullen, 1987). One of the first studies to examine this relationship was that of Shutts (1983), where the Eddy-

Straining Mechanism (ESM) was developed. In the ESM theory, diffluent flow immediately upstream of a dipolar high-over-low block causes transient synoptic eddies to become stretched and split into a poleward and equatorward component. Then, on the poleward side of the block, anticyclonic vorticity forcing induced by the eddies reinforces the blocking high; meanwhile the blocking low is maintained by the cyclonic vorticity forcing of the eddies on the equatorward side. The ESM can also explain how blocks are prevented from downstream advection by the background westerlies (Mullen, 1987), but does not consider how

eddies of opposing polarities interact with the block. The ESM also only assumes a meridional dipolar structure (akin to a Rex block) and therefore cannot be applied to all blocking events.

Yamazaki and Itoh (2009, 2013a, b) proposed a more general process for block maintenance called the Selective Absorption Mechanism (SAM), which is valid for blocks of all shapes, in all locations and at all times of year. The SAM assumes that blocks are large-scale areas of anticyclonic (AC) potential vorticity (PV) that attract smaller synoptic-scale anticyclonic eddies and repel synoptic-scale cyclonic eddies through differential vorticity advection. The attraction and absorption of the negative PV anomaly associated with the anticyclonic eddies reinforces the block, allowing it to persist for longer. The SAM also explains the reinforcement of cyclonic systems associated with omega or dipole blocking, if present, as these selectively attract synoptic lows and repel synoptic highs. This concept is expanded further in Luo (2005) and Luo et al. (2014, 2019) using the Eddy-Block Matching (EBM) mechanism theory. In the EBM mechanism, the two-way relationship between blocks and synoptic-scale eddies is explained via eddy vorticity forcing (EVF). If the background EVF is favourable for block amplification, the block feeds back onto the eddies to strengthen them and the background EVF, which further amplifies the block, and so on. Therefore, the presence of transient eddies is crucial to the maintenance of a blocking event, though there has been very little work to date on the exact quantitative relationship between eddies and blocks. Specifically, a study into the amount to which AC synoptic eddies are absorbed by blocks, or the magnitude of the eddies that contribute towards blocking, and how these eddy characteristics affect the persistence of a block is lacking.

All studies of blocking require some form of objective blocking definition. A vast selection of techniques have previously been proposed which include weather regime classification (Vautard, 1990; Grams et al., 2017) or self-organising maps (Thomas et al., 2021); field reversals in meridional geopotential height (Lejenäs and Økland, 1983; Tibaldi and Molteni, 1990; Scherrer et al., 2006) and potential temperature on the dynamical tropopause (Pelly and Hoskins, 2003; Tyrlis and Hoskins, 2008); anomalies in geopotential height (Charney et al., 1981; Shukla and Mo, 1983; Liu et al., 2018; Schiemann et al., 2020) or PV (Schwierz et al., 2004); and Rossby wave breaking (Masato et al., 2012; Shi and Nakamura, 2021). The diversity in blocking metrics arises from the fact that there is no perfect way to detect blocking events, and each method has been designed with a specific purpose in mind. The most common blocking indices have well-documented strengths and drawbacks, as highlighted in Barriopedro et al. (2010). A method that detects both blocks and their contributing anticyclonic transient features has not yet been developed and thus another detection method that is able to capture both is required.

A recent study by Okajima et al. (2021) classified AC and cyclonic synoptic eddies using a method based on the curvature vorticity. However, a more common way to detect and follow meteorological features is through objective feature tracking. This approach is common when studying cyclonic systems (e.g., Hodges et al., 2011; Catto et al., 2011; Sainsbury et al., 2020; Priestley et al., 2020), and a few studies have employed similar techniques for anticyclones (e.g., winter anticyclones affecting China in Chen et al. (2014); NH winter anticyclones in Ioannidou and Yau (2008); a global anticyclone climatology in Pepler et al. (2019)). Recent work by Liu et al. (2018) performed feature tracking on persistent 500 hPa geopotential height (Z500) anomalies to build a climatology of where these persistent anomalies (analogous to blocks) occur. However, these previous studies all focused on large scale anticyclones or blocks themselves, rather than the smaller AC eddies that help form and maintain them. This current study aims to identify and track these AC eddies that contribute towards blocking. Once identified, the relationship between transient AC eddies and block persistence will be established in terms of both (i) how the number of

eddies contributing towards a block affects its persistence, and (ii) how eddy intensity, size, and speed affects block persistence. This analysis will be performed for both the Euro-Atlantic and North Pacific regions, with a focus on winter and summer.

The paper is structured as follows. Section 2 describes the data used in this work, and also discusses how AC eddies and blocks are defined in this study. Section 3 demonstrates how AC eddies can interact with a block event with a short case study, and explains how AC eddies can increase the persistence of a block. Section 4 describes the spatial distribution, and persistence distribution, of blocks. The statistical relationship between block persistence and AC eddies is discussed in Sect. 5, first in terms of the number of AC eddies contributing to blocks, and second the strength and speed of the AC eddies. These findings are then explained with some dynamical arguments in Sect. 6. Finally, the work is concluded and summarised in Sect. 7.

## 2 Data and Methods

### 2.1 Data

The data used in this study are taken from the European Centre for Medium-Range Weather Forecasts (ECMWF) 5th generation reanalysis (ERA5) (Hersbach et al., 2020). The blocking index and feature-tracking are based on analysis of 6-hourly Z500 data from 1 March 1979 – November 2021 with an F128 grid resolution. This is a regular Gaussian grid, with a grid size of approximately $0.7°$, and is coarser than the full ERA5 resolution of $0.25°$. Experiments were performed at different resolutions and the conclusions made did not change appreciably. Thus, the F128 resolution was chosen in the interest of computational speed. Data are additionally separated into the traditional meteorological seasons for further analysis in this work: winter (December, January, February; DJF), spring (March, April, May; MAM), summer (June, July, August; JJA), and autumn (September, October, November; SON).

### 2.2 Anticyclonic Eddy Definition

#### 2.2.1 Anticyclonic Anomalies

Despite the large array of existing block detection methods (discussed in Sect. 1), the majority have some drawbacks that would make a climatological study of the AC eddies that contribute to blocking difficult. For example, some methods require a certain aspect of subjectivity, produce unrealistically small blocks, or even fail to detect blocks of a certain shape (Barriopedro et al., 2010). These issues are particularly prominent for Z500 reversal-based techniques. It is also desirable to identify the AC eddies and blocks at the same time, and mobile synoptic-scale AC eddies are unlikely to produce a marked reversal in the meridional Z500 gradient. Thus, a Z500 anomaly-based detection method was pursued.

In this study, blocks and AC eddies are defined as regions with a large positive Z500 anomaly from the zonal mean. The algorithm from Liu et al. (2018) is adapted in this study, to also consider the climatological wave patterns. The Z500 anomalies used to define the transients and the blocks ($Z'_*$) are calculated at each grid point (with longitude $\lambda$ and latitude $\phi$), at each time

step $t$, and are given by:

$$Z'_*(\lambda, \phi, t) = Z_*(\lambda, \phi, t) - \overline{Z_*}(\lambda, \phi, t) \tag{1}$$

where $Z_*(\lambda, \phi, t)$ is the instantaneous Z500 anomaly from the instantaneous zonal mean, and $\overline{Z_*}(\lambda, \phi, t)$ is the climatological (1979-2021) monthly deviation from the zonal mean Z500, where three-month smoothing has been applied.

At this stage, the importance of accounting for the climatological wave pattern ($\overline{Z_*}$) is noted. Figure 1 shows the monthly
climatological $\overline{Z_*}$ pattern for all 12 months of the year. A stationary wave train is evident for much of the autumn, winter, and spring months, with the largest anomalies in winter. These anomalies are also consistent with the shape of the climatological North Atlantic and North Pacific storm tracks (Hoskins and Hodges, 2019). Climatological ridging occurs over a large band from the central North Atlantic to central Eurasia, with a deep trough to the east of this centred over northern Japan. Over the eastern North Pacific and western North America, a smaller and less intense region of ridging is present, followed by
another trough downstream over the northeast of the continent. In summer, the same pattern manifests but is much weaker. Without considering $\overline{Z_*}$ when calculating the blocking index, regions of climatological ridging would show a large positive blocking frequency bias due to the high $Z_*$. Similarly, block frequency bias would be largely negative over regions of climatological troughing. In the work that follows, blocks are therefore considered to be anomalous circulation patterns within the climatological ridges and troughs, rather than simple anomalies in the zonal flow pattern.

### 2.2.2  Anticyclonic Eddy Tracking

The 6-hourly positive $Z'_*$ centres are followed using an objective feature-based tracking algorithm (TRACK; Hodges (1994, 1995, 1999)). Tracking begins when $Z'_* \geq 60$ m and stops when the strength of an eddy goes below this value. This low threshold allows for the path of the eddy to be tracked both before and after it is part of a block, providing insights into the life cycle of AC transient eddies that contribute towards blocks. The tracks of the eddies are then filtered according to whether they are considered to
contribute to a blocking event or not (Sect. 2.4). Further details about the configuration of TRACK used in this study are given in Sect. A1.

### 2.3  Blocking Index and Sector Blocking Definition

The $Z'_*$ field is also used to calculate a Eulerian blocking index at each grid point every six hours. For a grid point to be blocked, $Z'_*$ must exceed 100 m for five or more consecutive days. This results in a 43-year time series at each grid point determining the
periods in which the $Z'_*$ magnitude and persistence threshold are met for blocking. Finally, sector blocking events (hereafter "blocks") are defined to occur when the area of blocked grid points inside a domain exceeds $1.0 \times 10^6$ km$^2$ (around 10% of either domain). This is only half the minimum size criterion imposed in Schiemann et al. (2020) for the ANOM index, but sensitivity tests showed that the results presented in this study are not dependent on the minimum size threshold (not shown). It should be noted that while a 5-day persistence criterion is imposed on the grid point-level $Z'_*$ index, a persistence threshold
is not applied to the definition of sector blocking. It is therefore possible that a sector blocking event lasts for fewer than five

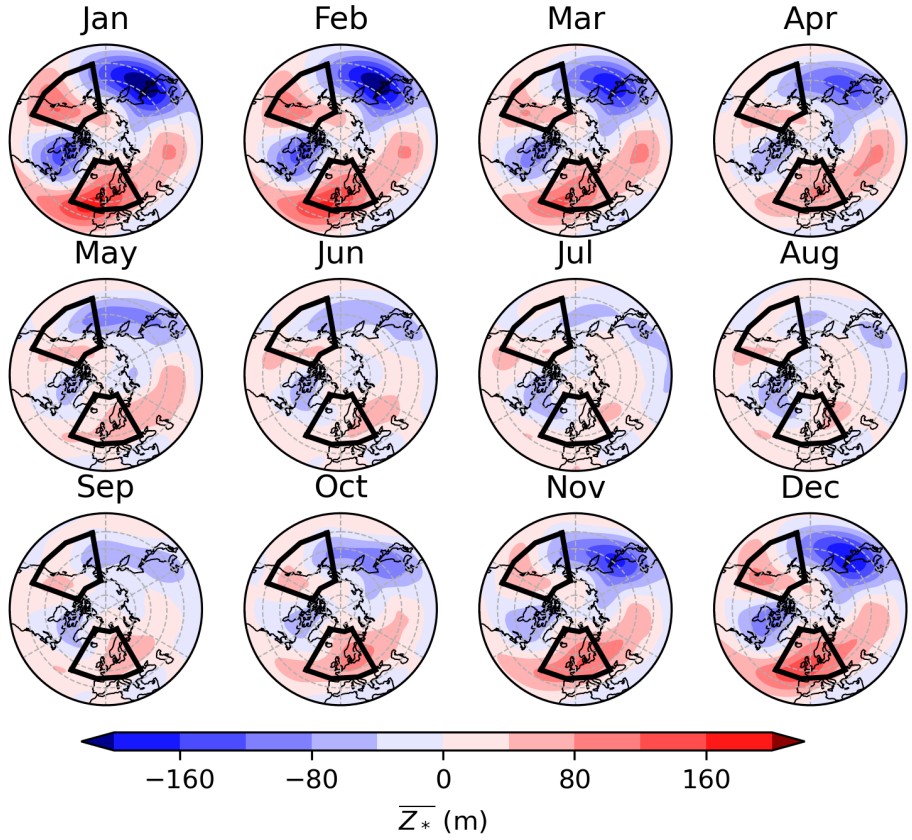

**Figure 1.** Monthly climatological $\overline{Z_*}$ from 1979–2021 (in metres) for January–December (shading). Also shown are the ATL and PAC regions (black boxes) used in this study (defined in Sect. 2.3).

days; these "edge cases" represent blocks that occur near the boundary of the domain, or a smaller anticyclonic event in the domain that fluctuates around the area threshold in size. More details of this scenario are given in Sect. A2.

Sector blocking events are determined for two regions: the Euro-Atlantic (hereafter "ATL", 30° W–30° E, 45° N–75° N) and the North Pacific/northwest North America (hereafter "PAC", 170° W–110° W, 40° N–70° N). The domains are shown by black boxes in Fig. 1. These domains are both 60 degrees longitude wide, which is similar to the domain width used in Pelly and Hoskins (2003) but 15 degrees larger than in Tyrlis and Hoskins (2008). However, the ATL and PAC domains were defined to be this wide in this study to account for the moving position of the climatological blocking maxima in these regions in different seasons. This was done while also minimising the chance that more than one blocking event is captured in the domain at the same time. The regions were also designed to align with large population centres with relatively large climatological blocking frequencies, such that the blocks analysed in this study have the potential to cause widespread impacts. For example, the PAC domain as defined in this study is able to capture the dynamics from the severe North American heatwave in June

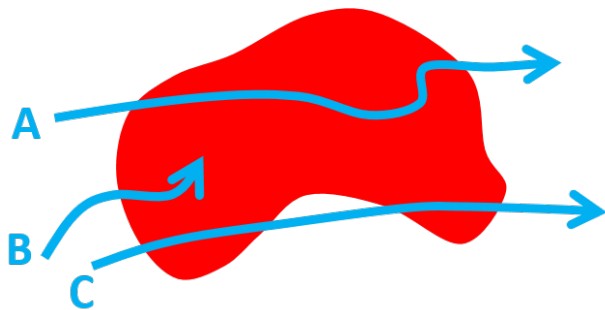

**Figure 2.** The three scenarios where AC tracks are considered to be contributing to blocking. Track A shows an eddy that passes through the block, Track B shows an eddy absorbed by the block, and Track C shows an eddy that fluctuates between coinciding with a block and not with a block. Tracks A, B and C are all considered to be AC eddies contributing to blocking.

2021, which would not be the case if the domain was positioned closer towards the climatological summer blocking frequency maximum.

### 2.4  Attributing Eddy Tracks to Blocks

A final step is required to associate $Z'_*$ tracks with blocking as defined by the blocking index for further analysis. Three scenarios where the tracks and blocks overlap are considered to be contributing eddies (Fig. 2):

- "Through" eddies (Track A in Fig. 2), where a $Z'_*$ track starts and finishes outside a block, but travels through a group of blocked grid points somewhere in its lifetime.

- "Absorbed" eddies (Track B) where a $Z'_*$ track starts outside and finishes inside a collection of blocked grid points.

- "Edge" eddies (Track C) that fluctuate between coinciding with blocked grid points, and outside a block. These tend to occur on the edge of a group of blocked grid points, or in the onset/decay phase of a block.

Anomaly tracks can also coincide in space and time with blocking events in two other ways, which will briefly be mentioned here. First, "internal" tracks are $Z'_*$ tracks that predominantly (over 80% of the time) remain inside a block throughout their lifetime. These are normally very slow-moving features, which are representative of the movement of the blocking anticyclone

centre. These tracks are therefore not considered to be AC eddies that contribute to blocking. Secondly, blocks can also occasionally produce "spawned" eddies, which are $Z'_*$ tracks that start inside a blocked region and leave at some stage later in their lifetime. These cases are also not considered to be eddies that contribute to the block they spawn from, however they may go on to become an AC eddy associated with a different blocking event outside of their genesis region.

## 3 Case Study of a Block and its Transient Eddies

In this section, a case study is discussed to illustrate how transient eddies can contribute to both establishing and maintaining a blocking event. This event meets the sector blocking definition in the ATL region for 11.25 days, from 25 February–8 March 2011, and has a total of two AC eddies (with an additional two internal tracks and one spawned track). Figure 3 shows the $Z'_*$ field and blocked points at 12 UTC for every day of the event. Additionally, all AC eddy, internal, and spawned tracks for this event are shown.

On 25 February, a ridge breaking in the far east of the domain gives rise to grid-point level blocking (Fig. 3a). At this stage, there is no $Z'_*$ track in the domain, though one is present over the Scandinavian block just outside the ATL domain (not shown). By 26 February (Fig. 3b), a small AC eddy (dark blue line) originating from the mid-Atlantic can be seen approaching from the west as a low-amplitude ridge in the $Z'_*$ field. This eddy enters the domain on 27 February (Fig. 3c) and this associated ridge amplifies, while the block situated in the east of the sector over Scandinavia remains in place. On 28 February (Fig. 3d), the western ridge breaks and connects to the Scandinavian block, resulting in many more blocked grid points in the domain, even in the west by this stage. Also on 28 February, the ATL block spawns an eddy (light blue line in Fig. 3) that travels towards the Ural Mountains until 3 March. A small internal track is also present on the far eastern part of the domain, associated with a slight movement in the central anticyclone between 28 February 12 UTC and 1 March 12 UTC (so no yellow dot is shown). Meanwhile, another AC eddy has already begun travelling towards the block. This eddy orignated from the United States, and will eventually travel all the way to Japan, via the ATL block, travelling around 240 degrees longitude from start to end. At this early stage (28 February), this eddy is still fairly small, but over the next few days it grows in both size and amplitude as it connects to the ATL block (Fig. 3d-f). Another, more prominent internal track develops inside the block from 1–2 March. Once inside the block, the Atlantic AC eddy slows down (as shown by the loop in its track from 3–5 March, Fig. 3g-i). On 6 and 7 March, the AC eddy begins to speed up again as it travels east towards Japan, and the ATL block slowly decays (Fig. 3j-k). Once the track leaves the ATL domain, very few grid points remain blocked and the block event finishes.

A short discussion on how AC eddies contribute to blocking dynamics is now introduced. Figure 4 shows how the area and intensity of the ATL sector block introduced here varies throughout its lifetime, and also shows the timings of the AC eddy interactions with it. The ATL block area (black line) quickly increases in the first 3–4 days, remains relatively constant at $5.0 \times 10^6$ km$^2$ for 2–3 days, and then decreases again in the final 4–5 days. The ATL block intensity (grey line) starts off high, associated with the intense Scandinavian block to the east of the ATL domain. After a brief decrease, the intensity increases to over 350 m just after the area reaches its maximum. After 3 March, the intensity steadily decreases along with the block area in the ATL region.

The arrival of the first AC eddy into the block is associated with the sharp increase in sector block area, and the second AC eddy coincides with the rise in intensity of the block. The increase in block area (intensity) occurs 12–18 hours before the first (second) AC eddy centre enters the block, implying that the eddies have a field of influence that extends beyond their tracked centres. Increases in block intensity and/or area that coincide with the arrival of an AC eddy are also detected for many other blocks in this study (not shown). These findings are very important in determining the longevity of a block. Larger, more

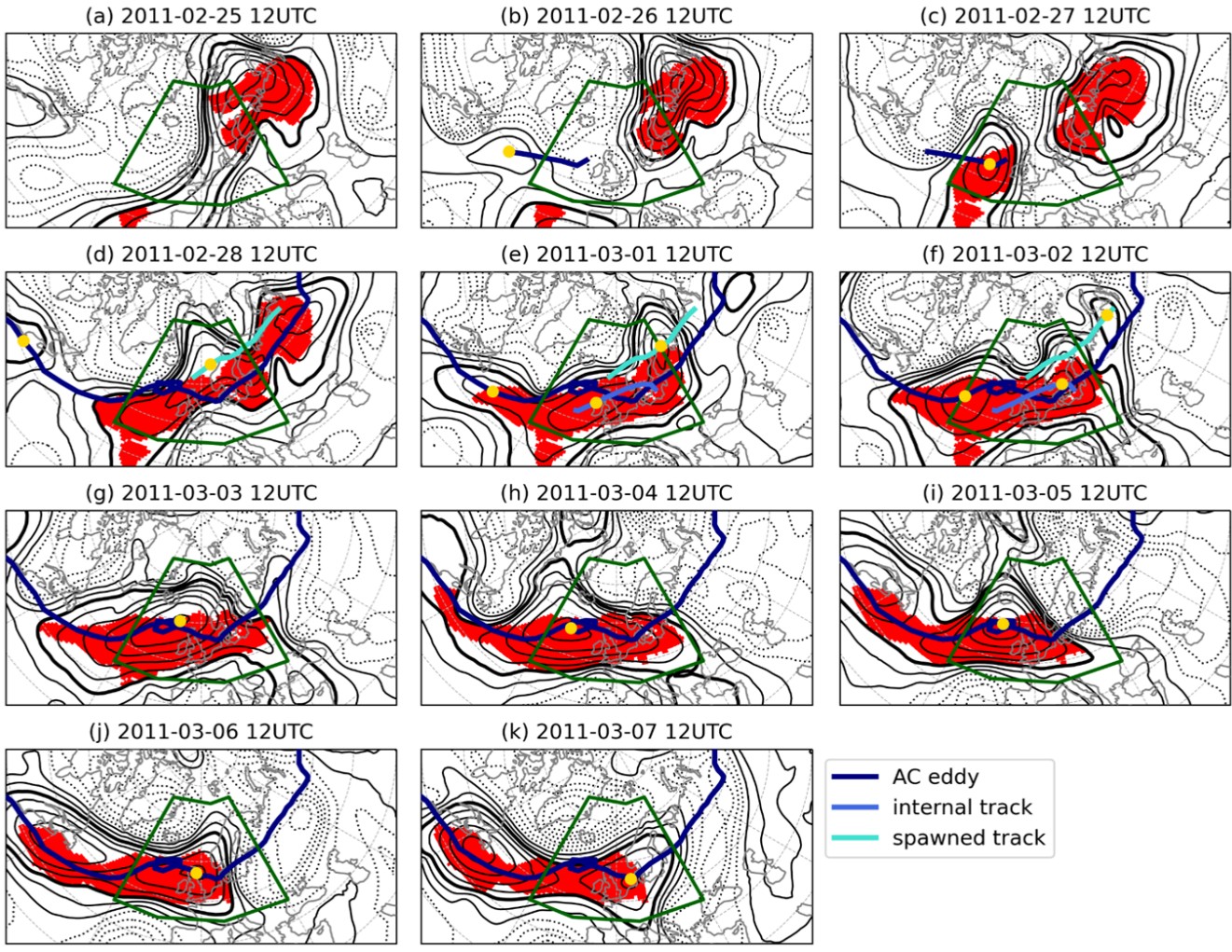

**Figure 3.** Blocked grid points (red shading) and $Z'_*$ anomaly field (black contours at 50 m intervals; negative values dashed, 100 m line bold) at 12 UTC for each day of an ATL blocking event (the ATL region is depicted by the green box). AC eddy tracks (dark blue), internal tracks (medium blue) and spawned tracks (light blue) that coincide with this blocking event are also shown, with each track's position at the valid time shown by the yellow dot.

intense blocks require more time to advect them downstream or dissipate (e.g., Yamazaki and Itoh, 2013b), and thus AC eddies that strengthen or broaden blocks can also be expected to increase their persistence.

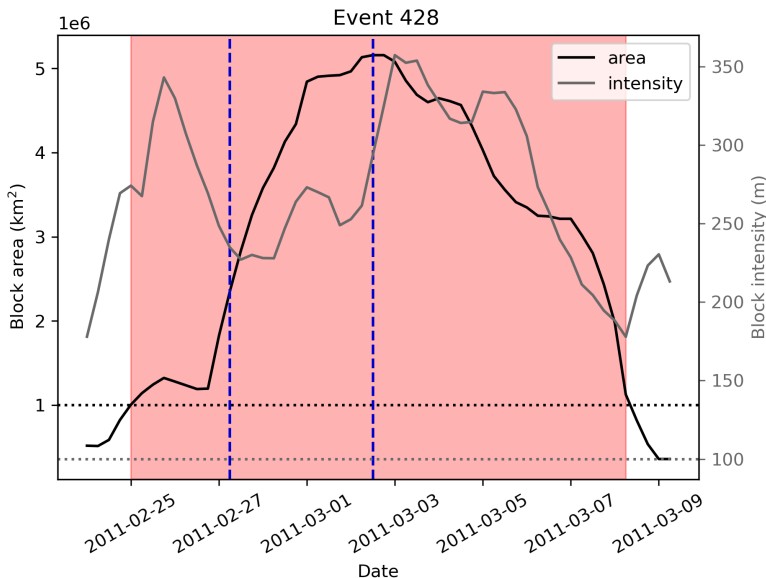

**Figure 4.** Time series of the blocking example introduced in Sect. 3, showing the block area (black, left axis) and intensity (grey, right axis; measured as the maximum $Z'_*$ in the ATL domain at each time). Red shading indicates the times at which sector blocking was occurring in the ATL domain, with horizontal dotted lines showing the minimum area and intensity thresholds for sector blocking. Vertical blue dashed lines denote the times at which AC eddy centres (yellow dots in Fig. 3) first coincide with a blocked grid point in the ATL domain.

## 4 Northern Hemisphere $Z'_*$ Index Blocking Climatology

Before examining the transients that maintain blocks in the ATL and PAC domains, a climatology of NH blocking using the $Z'_*$ index (from Eq. 1) is presented to motivate the selection of these two regions. Figure 5 shows the percentage of blocked days in winter, spring, summer, and autumn for the Northern Hemisphere. The spatial distribution of blocking frequency is consistent with many previous studies that utilise Z500 anomalies for block detection (e.g., Barriopedro et al., 2010; Schiemann et al., 2017; Woollings et al., 2018). Blocking occurs most frequently in three regions: the Northeast Pacific/Northwest North America, Northeast Atlantic/Western Europe, and Scandianvia/Ural Mountains. The effect of considering the climatological stationary wave features ($\overline{Z_*}$) is noticeable here by comparing Fig. 5 with that of Liu et al. (2018) where $\overline{Z_*}$ is not considered (their Fig. 9). Climatological blocking frequencies as measured using the $Z'_*$ index are half as frequent in the Pacific and Atlantic maxima than in Liu et al. (2018) (climatological frequency of 16% vs 30%). The seasonal variation in blocking frequency found in many previous studies (e.g. Dole and Gordon, 1983; Barriopedro et al., 2010; Schiemann et al., 2017; Woollings et al., 2018) is also reproduced here, with blocking being most common in winter and least common in summer.

While Fig. 5 implicitly shows how many days in each season are blocked on average, the length of each individual block can vary greatly around its average length. The distribution of sector block persistence for the ATL and PAC regions is shown in Fig. 6, along with the first (Q1), second (Q2), and third (Q3) quartiles. Results for winter and summer are discussed in detail

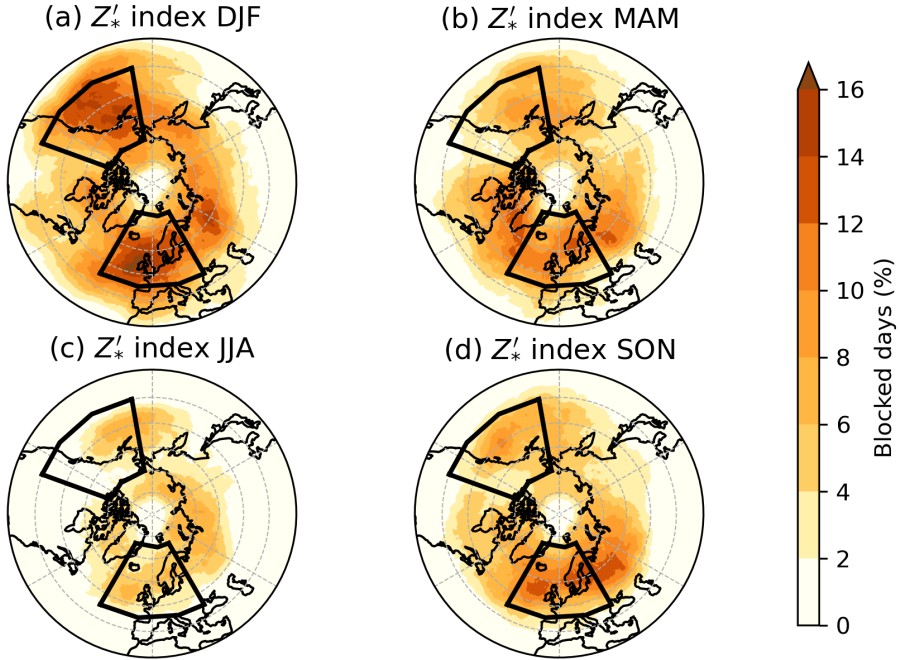

**Figure 5.** Blocked day frequency according to the $Z'_*$ index, showing the percentage of days in a season that are blocked in (a) winter, DJF; (b) spring, MAM; (c) summer, JJA; and (d) autumn, SON. Black boxes indicate the ATL and PAC regions.

here since these are the seasons where blocking has the potential to bring the most severe hazards, with spring and autumn distributions shown for completeness in Sect. A3. Sector block events with a persistence of fewer than 5 days are generally uncommon in DJF, but the JJA Q1 values are less than five days. This indicates that a sizeable proportion of JJA blocks are small in size and only marginally exceed the minimum area threshold for sector blocking.

The shape of the persistence distributions shown in Fig. 6 is qualitatively similar to the distributions found in other studies that use different blocking indices (e.g., Wiedenmann et al., 2002; Diao et al., 2006; Drouard and Woollings, 2018; Detring et al., 2020) in that shorter blocks are far more common than long events. The high climatological blocking frequency in winter shown in Fig. 5 is due to both a larger number of blocking events, and a longer duration of these blocks. In both sectors, the quartiles of block persistence are much larger in DJF than JJA, with DJF blocks having a median length comparable to the third quartile in JJA. The distribution of block persistence is remarkably similar in the ATL domain for DJF, MAM and SON (MAM and SON shown in Fig. A2), with the longest blocks lasting 39.25 (DJF) and 39.75 (SON) days, while the longest JJA ATL block persisted for 23 days. Slightly more seasonal variation with block persistence is found in the PAC region (c.f. Fig.

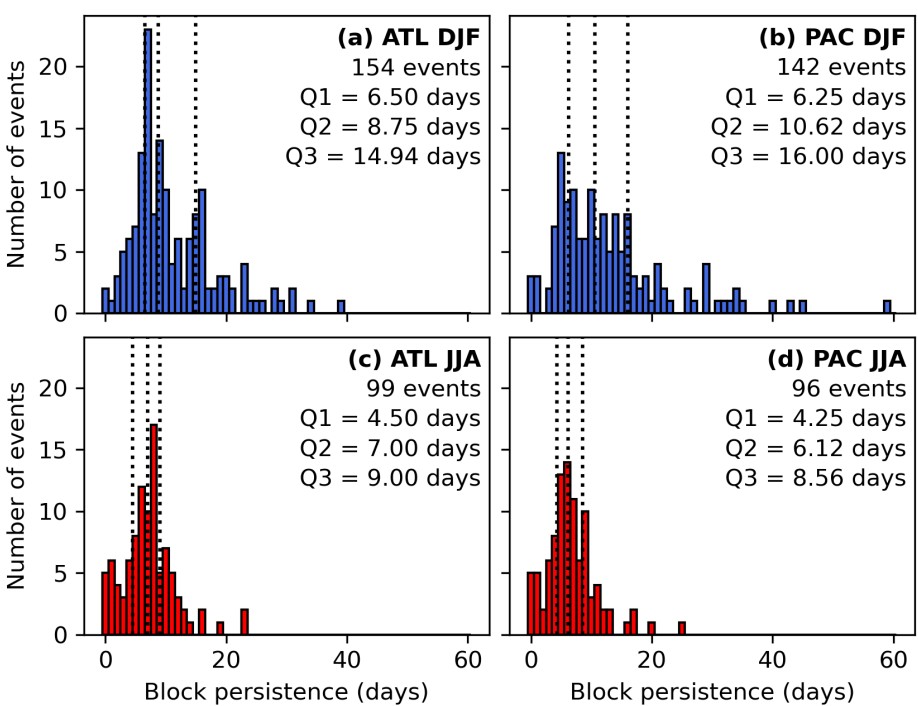

**Figure 6.** Histograms showing sector blocking event persistence frequency in winter (a, b) and summer (c, d) for the ATL (a, c) and PAC (b, d) sectors. The first (Q1), second (Q2), and third (Q3) quartiles of block persistence are indicated by the dotted lines.

A2), and with the exception of DJF (where the longest block is 59 days long), PAC block lifetimes are slightly shorter than those found in the ATL domain.

# 5   Relationship between Block Persistence AC Transient Eddies

## 5.1   Number of AC Eddies

The persistence, mean area, and number of associated transient AC eddy tracks of each blocking event for both regions in winter and summer are shown in Fig. 7. Correlations between these three measures are also shown on each panel, and all are statistically significant ($p < 0.05$). As with the histograms in Fig. 6, MAM and SON blocks behave similarly to DJF blocks in terms of AC eddy number, thus only winter and summer blocks are discussed here, with results from MAM and SON included in Sect. A4.

Longer blocks are generally larger, and this relationship is strongest in summer in the PAC region (correlation of 0.71). Larger blocks take longer to naturally dissipate (Yamazaki and Itoh, 2013a), so this result is not surprising. Additionally,

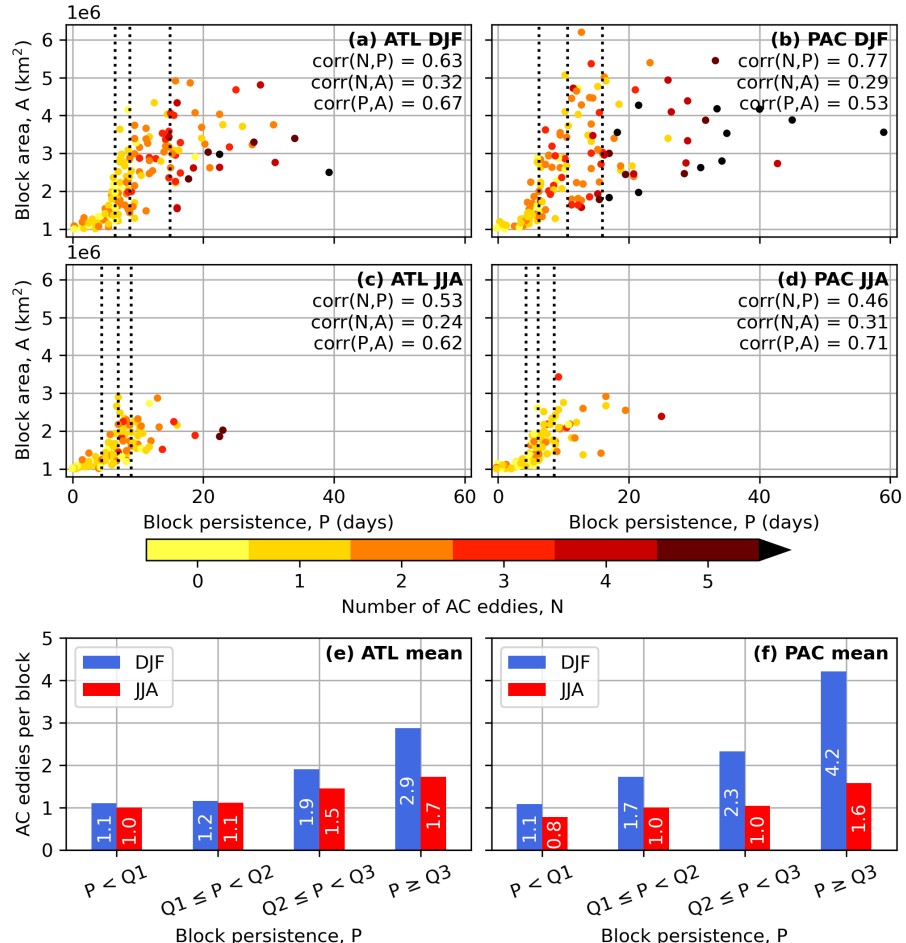

**Figure 7. (a-d)** Number of anticyclonic transient eddies (N) that contribute to blocking events in winter (a, b) and summer (c, d) for the ATL (a, c) and PAC (b, d) regions. Blocking events are characterised by their persistence (P) and mean block area (A). Pearson's correlation coefficients between N, P, and A are indicated for each sector and season (all are significant to the 95% confidence interval). Q1, Q2, and Q3 (as in Fig. 6) are indicated by the dotted lines. **(e-f)** Mean number of AC eddies per block of a particular persistence, defined in terms of the quartiles for a season, for the ATL (e) and PAC (f) regions in DJF (blue) and JJA (red).

more persistent blocks generally have more transient AC eddies contributing to them, and this is a combination of a larger number of these eddies either ending in or passing through the block. The correlation between block persistence and block

255 area is comparable to the correlation between persistence and number of AC eddy interactions in DJF for both sectors. The JJA correlations between persistence and number of AC eddies are weaker but still reasonably strong, suggesting that the number

of AC eddy interactions with a block is somewhat less important in summer. The relationship between the area of a block and the number of AC eddies it interacts with is comfortably the smallest of the three correlations shown, as for a given area, there is large variability in the number of AC eddies a block interacts with.

Despite the general observation where more persistent blocks interact with more AC eddies, there is also substantial variability in the number of AC eddies for a block of given persistence, where between 2 and 6 eddies can be seen to be contributing the most persistent 25% of blocks (Fig. 7a-d). However despite this large variability, the number of AC eddies a block interacts with increases as its persistence increases at all times of year (Fig. 7e-f). The shortest 25% (P < Q1) of winter ATL blocks have an average of 1.1 AC eddies contributing to them, but the longest 25% (P ≥ Q3) have nearly three times as many eddies (2.9). The relationship is even stronger in the PAC sector, where the longest 25% of blocks interact with nearly four times as many AC eddies as the shortest 25% of blocks (4.2 and 1.1 eddies, respectively). The aforementioned 59-day DJF PAC block event interacted with eight upstream AC eddies. In summer in both sectors, the longest 25% of blocks only have up to twice as many AC eddies contributing than the shortest 25%, confirming that the relationship between AC eddy number and persistence is weaker. While corr(N,P) in JJA is smaller than in DJF for both sectors (0.53 vs 0.63 in ATL, 0.46 vs 0.77 in PAC), JJA blocks are also less persistent. Therefore, it is perhaps not surprising that the average number of AC eddies per block for longer blocks is smaller in JJA than DJF. Correlations between N, A, and P in both sectors in MAM and SON are generally comparable to those in DJF, with an average number of AC eddies per block larger than JJA but smaller than DJF (Fig. A3).

## 5.2 AC Eddy Strength and Speed

It is possible that the characteristics, in addition to the amount, of AC eddies interacting with a block can influence its persistence. First, the mean strength (as measured by the magnitude of $Z'_*$ at the AC eddy centre), and zonal and meridional speeds of all AC eddies are discussed, for blocks of all persistences. Black lines in Figs. 8, 9 show the strength, zonal speed and meridional speed of all AC eddies in the 7 days before and after entering blocks in DJF or JJA respectively, in both the ATL and PAC domains. The general behaviour of the eddies is similar in both the ATL and PAC regions, and the charactersitics of "absorbed" and "through" eddies are qualitatively similar to those of all AC eddies (not shown). In the days before the eddies enter a block, their strength remains fairly constant in JJA, and in DJF for ATL sector blocks (Figs. 8a, 9a-b). AC eddies contributing to DJF ATL blocking are stronger than their PAC counterparts before entering the block. JJA eddies are of a similar strength in both domains, but weaker than those in DJF. In the ATL domain, AC eddies in both DJF and JJA begin to strengthen by day -1, and then strengthen further by around 50 m in the first two days after entering a block (time = 0 line in Figs. 8, 9). In the PAC region, the intensification is stronger for DJF eddies (nearly 100 m) and begins at around day -3. It is possible that this could be a sign that the block is acting to strengthen the upstream AC eddies, consistent with the EBM mechanism (Luo, 2005). However, it could also potentially because the PAC region is slightly to the east of the North Pacific climatological blocking maximum (Fig. 5), meaning AC eddies intensify upon entering a block just outside the domain. However, further intensification does occur once the eddies are inside a block in the PAC domain between days 0 and +1. After this, in both regions, there is a steady decay of the eddy strength (though they are still as intense or stronger than they were before blocking). However, the strength

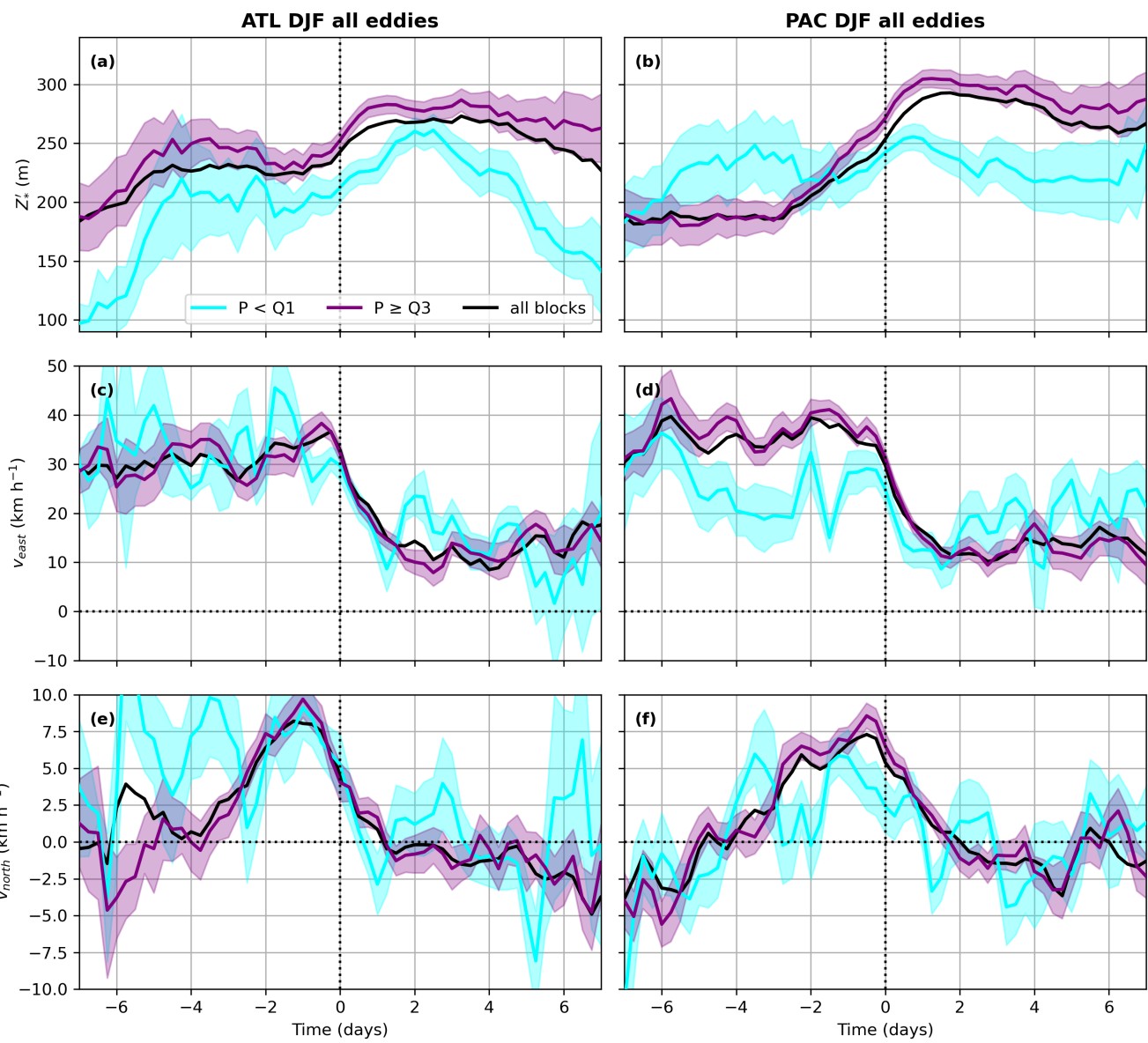

**Figure 8.** Characteristics of transient AC eddies that contribute to blocking in DJF. The mean strength, measured by the maximum $Z'_*$ (a, b), zonal velocity $v_{east}$ (c, d), and meridional velocity $v_{north}$ (e, f) of AC eddies that contribute to blocks of all lengths in the ATL (left) and PAC (right) domains are shown in black. The same mean quantities are also shown for only the AC eddies that contribute to the shortest (cyan) and longest (purple) 25% of blocks, with shading indicating the standard error from the mean at each time step. Negative times indicate the period before the eddy enters a block in the domain, and positive times indicate times after entering a block.

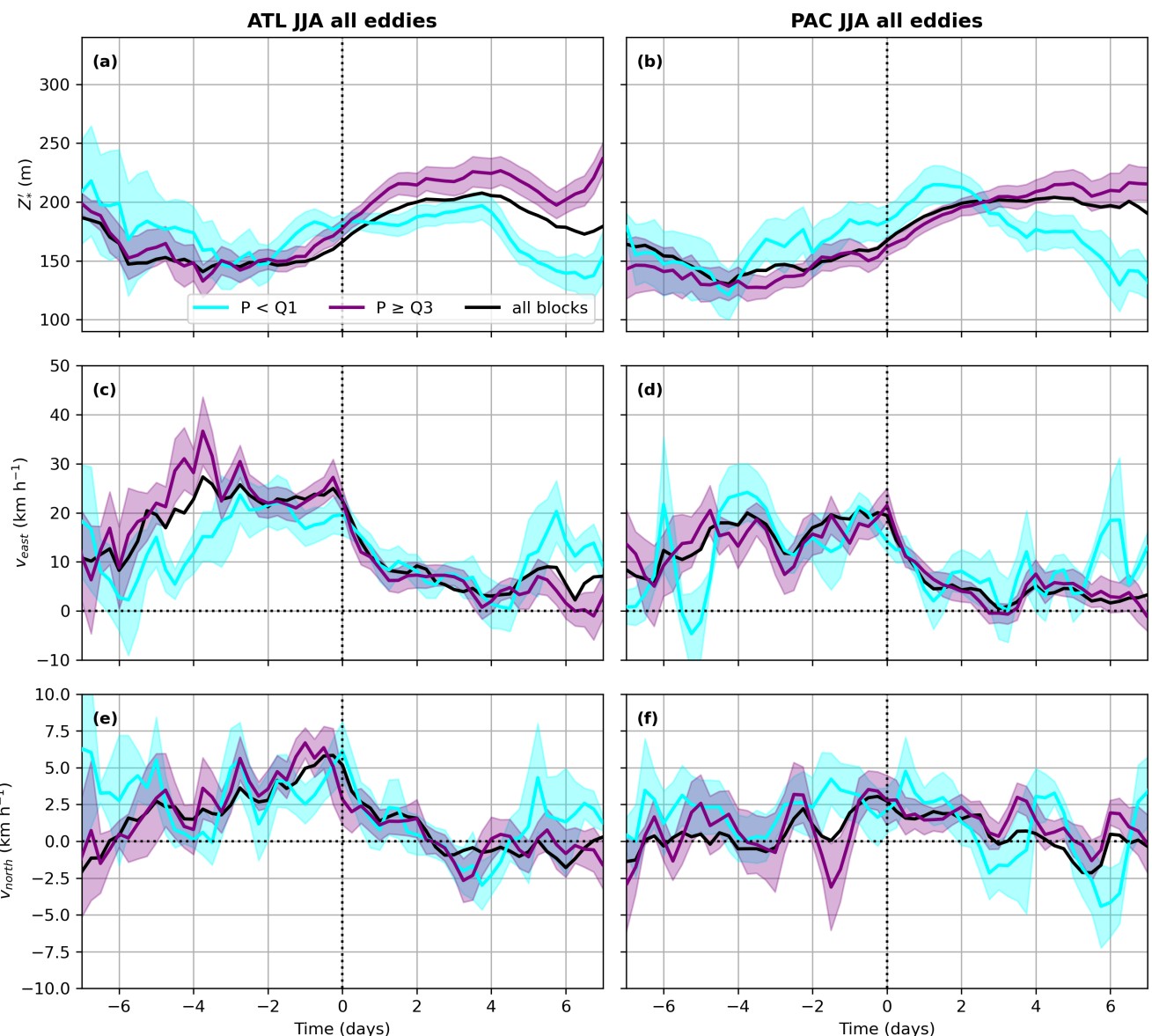

**Figure 9.** As Fig. 8, but for AC eddies that contribute to JJA blocks.

of a block can be maintained by the absorption of additional AC eddies later in its lifetime (not shown), if it persists for long enough.

The AC eddy zonal speed contributing to all blocks (black lines in Figs. 8c-d, 9c-d) is also fairly constant in the days before the eddy enters a block. Winter eddies move eastwards faster than those in summer, and the difference between seasons is largest in PAC. The slower eastward progression of AC eddies in JJA is consistent with the slower and weaker jet streams

during summer months. Despite the speed differential of the eddies between DJF and JJA, it is found that the distance travelled by each AC eddy does not change considerably between the two seasons (not shown). Upon entering the block, the eddies rapidly decelerate such that their zonal speed halves compared to days -7 to 0. Unlike with eddy strength, the zonal speed of the eddies remains constant through to day +7 and beyond.

The evolution of the meridional speed of the AC eddies that contribute to all blocks in both domains is more complex (black lines in Figs. 8e-f; 9e-f). In the seven days before entering a block, AC eddies gradually accelerate northwards in both domains during winter, and in summer in the ATL, though their meridional speeds are around four times smaller than their zonal speeds. This northward acceleration is potentially indicative of two things. Firstly, it signifies the building of a ridge through the northward advection of higher geopotential heights, which is a characteristic of blocking. However, it can also be the result of eddies being attracted by the block via the SAM. AC eddies then rapidly decelerate meridionally once inside a block, reaching small southward speeds. The rapid deceleration of the eddies, both zonally and meridionally, is typical of blocking dynamics, since blocking systems are quasi-stationary. Meridional speed for PAC summer blocking AC eddies is fairly constant before and during blocking. In MAM and SON (Sect. A4), AC eddies behave in largely the same way as outlined here, with intermediate speeds and strengths between those in DJF and JJA.

A similar analysis can be performed for AC eddies that interact with the shortest and longest 25% of blocks in both sectors (cyan and purple lines respectively in Figs. 8, 9), to determine whether there is a relationship between the strength or speed of the eddies and the persistence of the blocks they contribute to. If there is no overlap of the standard errors from the mean between AC eddy characteristics of different block persistences, then the results are said to be statistically significant.

The longest 25% of DJF ATL blocks interact with stronger AC eddies than the least persistent 25% of blocks (Fig. 8a), with the mean eddy strength for longest blocks being about 40 m larger at the time that the eddies enter the block. The same is true in the PAC domain in winter (Fig. 8b). The longest 25% of blocks in both sectors in SON also absorb stronger AC eddies than the shortest 25% of blocks (Fig. A5a-b). In both domains during JJA, the longest blocks are not the result of interacting with stronger AC eddies (Fig. 9a-b), and in fact stronger AC eddies result in less persistent blocks in the PAC in JJA. Additionally, the AC eddies contributing to both the longest and shortest 25% of blocks at t = 0 in the ATL region in JJA are stronger than the mean eddy strength, perhaps suggesting two contrasting pathways for block maintenance. The persistence of MAM blocks in the ATL domain also does not appear to depend on the strength of the absorbed AC eddies, but in the PAC similar behaviour to that in DJF and SON is shown (Fig A4).

It may appear from Figs. 8a-b, 9a-b that AC eddies that interact with the least persistent blocks become weaker and fast-moving again from day +4 onwards. To some extent, the same is true at times over +7 days for AC eddies that contribute to more persistent blocks (not shown). While it is true that such AC eddies do weaken and speed up at these times, the block is no longer present at these time steps, so these eddies are no longer interacting with blocks.

AC eddies contributing to the longest PAC blocks intensify by about 120 m from days -3 to +1, which is double the amount seen in the ATL domain in DJF. Associated with this marked increase in intensity, these AC eddies also move significantly faster than eddies contributing to shorter blocks (Fig. 8d, f). This speed discrepancy at the point of absorption is only also demonstrated in MAM (Fig. A4), which suggests that AC eddy speed is not usually a reliable indicator of how long a block

330 may persist for. In both domains in JJA (and MAM, Fig. A4), AC eddies that contribute to the shortest 25% of blocks do not markedly intensify upon entering a block like those entering longer blocks do, underlying the more transient nature of these shorter-lived block events. It is therefore possible that the lack of intensification of AC eddies upon entering a block could be a signal that the block will not persist.

## 6 Discussion

335 Yamazaki and Itoh (2013a) explain that blocks require a source of low-PV in order to counteract the effects of dissipation, and AC eddies can be thought of as a way to replenish the low-PV inside the block. Although the AC eddies in this study are defined using Z500, the invertibility principle of PV (Hoskins et al., 1985) means that ridges of low-PV correspond to ridges of high Z500, and therefore similar arguments can be made here. Thus, eddies that bring an AC Z500 anomaly towards a block can also be expected to provide an AC anomaly of PV, so we can therefore approximate both blocks and AC eddies to be vortices of 340 low-PV. Following the SAM, blocking vortices induce a stronger AC forcing than the eddies, which results in the eddies being attracted towards the block. When AC eddies enter a block, vortex merging occurs, where the two vorticity centres combine into a larger, stronger vortex. This new blocking vortex then exerts a stronger AC vorticity on its surroundings than before, which allows AC eddies to be attracted towards the block from further away, until they themselves merge with the block. This positive feedback loop potentially allows blocks to become self-sustaining (Yamazaki and Itoh, 2013a), and therefore leads to 345 some very long blocking events with many contributing AC eddies (like those seen in DJF in the PAC region in Fig. 7).

As shown in Sect. 5.2, block persistence in the ATL domain is less sensitive to AC eddy strength than in the PAC region, particularly in MAM and JJA. The correlation between the number of AC eddies and block persistence is also weaker (though still relatively strong) in the ATL than PAC (Sect. 5.1). It is possible that in the ATL region, competing blocking dynamics are being detected for the Atlantic and continental European portions of the domain. Miller and Wang (2022) showed that synoptic-350 scale fluctuations in Z500 are important factors in European blocking dynamics, whereas Atlantic block dynamics are instead determined more by planetary-scale, longer timescale anomaly patterns such as the North Atlantic Oscillation (NAO). These differing factors may account for the weaker dependence of ATL block persistence on the number and strength of AC eddies.

The finding that longer JJA PAC blocks result from the absorption of weaker AC eddies results from the positioning of the PAC domain. In JJA, the PAC domain only partially covers the climatological blocking frequency maximum in this area (Fig. 355 5), with a large portion to the west not being considered in this study. When the PAC domain is shifted 30°W to cover the entire climatological blocking maximum (not shown), no statistical significance exists between eddy strength for the shortest and longest 25% of blocks, akin to the ATL in JJA (Fig. 9a). Therefore, it can also be deduced that counteracting blocking dynamics are being detected in the original PAC domain. The strongest AC eddies lead to more blocking to the west of the PAC domain, resulting in only small persistences in the PAC due to it only capturing the eastern flank of such block events. Weaker 360 AC eddies lead to more blocks inside the PAC domain itself, leading to higher persistences since the blocks are wholly within the sector. This explains why, for the PAC domain in JJA, it appears that longer blocks interact with weaker AC eddies. Clearer relationships between AC eddy strength and block persistence could be produced when the PAC (and to a lesser extent, the

ATL domain) are aligned more with the seasonal climatological block frequency maxima, however the results presented here are still important since the blocks analysed have the potential to cause more impacts than those e.g. over the Pacific Ocean. The methodology presented here can be applied robustly anywhere, provided that climatological block frequency is relatively high.

## 7   Conclusions

This study has used objective feature-tracking of synoptic-scale AC eddies that help contribute to atmospheric blocking events to analyse the climatological relationship between transient AC eddy number, intensity, and block persistence in the North Pacific (PAC) and Euro-Atlantic (ATL) regions. It is found that in both sectors, more persistent blocks are associated with more transient AC eddies, and this relationship is weaker in summer compared to other times of the year. The PAC region exhibits a larger variability in the number of eddy interactions for blocks of different lengths than the ATL region, though both regions show that the most persistent blocks interact with the most AC eddies. These results suggest that blocks can be maintained through repeated absorption of AC eddies, potentially supporting the SAM theory (Yamazaki and Itoh, 2013a). In general, the number of AC eddies a block interacts with is important for determining its persistence in all locations at all times of year. However, not all persistent blocks are the result of a large number of AC eddy interactions, which indicates that other dynamical processes are also important for block maintenance in these cases (e.g., interactions with waves originating from the tropics (Austin, 1980); and moist dynamics (e.g. Pfahl et al., 2015; Steinfeld and Pfahl, 2019)).

Conversely, the relationship between the persistence of blocks and the strength of the AC eddies it absorbs is more complex. At all times of year, block length in the PAC region is also dependent on AC eddy strength. In all seasons apart from summer, the most persistent 25% of blocks have absorbed statistically significant stronger AC eddies than the least persistent 25% of blocks. In summer the reverse is true in that longer blocks result from the absorption of weaker AC eddies, though this is perhaps an artefact of the choice of domain. For the ATL domain, stronger AC eddies only increase the length of the block in autumn and winter, whereas ATL block length appears to be unaffected by eddy strength in spring and summer. Therefore, the relationship between block persistence and AC eddy strength appears to be more variable than that with the number of AC eddies, due to the dependence on location and time of year.

AC eddies increase the persistence of blocks through increasing their area and/or intensity. This means that blocks require a longer period of time to either advect them downstream, or naturally decay through dissipation. A larger number of, or more intense, AC eddies result in larger block area or intensity increases, thus leading to longer-lasting blocks.

Analysis of AC transient eddies associated with blocks of all lengths in both sectors leads us to conclude that winter eddies are stronger and faster than their summer counterparts. Most AC eddies intensify and accelerate northwards towards the block just before entering, which could potentially signal an attraction via the SAM, (Yamazaki and Itoh, 2013a). Nonetheless, more evidence on the vortex-vortex interactions between AC eddies and blocks are required to ascertain whether this mechanism is actually taking place. However, generally AC eddies that enter the least persistent 25% of blocks do not undergo this intensification once inside the block, thus meaning that this behaviour could be used as a potential indicator for how long a

block may persist for. All AC eddies rapidly decelerate once inside a block, consistent with the slow-moving nature of block events.

This study only considers dry dynamical processes, namely multi-scale interaction between the large-scale blocks and smaller-scale AC eddies. While the results presented here suggest there is a strong, significant relationship between block persistence and the amount of AC eddies a block interacts with, this process is certainly not the only dynamical process occurring during the maintenance phase of a block. The most important missing piece of this study is the extent to which moist dynamics, for example diabatically-heated outflow from warm conveyer belts (e.g. Pfahl et al., 2015; Steinfeld and Pfahl, 2019), also affect block persistence. We hypothesise that in persistent blocks with very few AC eddy interactions, other maintenance processes such as diabatically-generated negative PV anomalies are instead dominant. Similarly, in short blocks with many AC eddy interactions, there may be other processes (e.g. diabatic cooling) that causes the block to decay quickly despite the continued AC eddy forcing. Further work is required to compare these dynamical differences between blocks with many contributing AC eddies to those with few AC eddy contributions. Furthermore, our results have also highlighted the existence of two further types of AC eddies, namely those that pass through the block, and those that are spawned by the block and propagate downstream. These types of AC eddies require further investigation, particularly as it is possible that they can go on to interact with another block event downstream. Finally, this work has only considered the AC eddies that contribute to blocking anticyclones, whereas some blocks (omega or dipole blocks) also have quasi-stationary cyclones as part of the blocking system. Further analysis is required to examine whether more (or more intense) cyclonic eddies increase the persistence of blocking cyclones, in a similar way that AC eddies increase the persistence of the anticyclonic part of blocks.

*Code and data availability.* The code used to obtain the results in this work are available from the authors upon request. The underlying data used for the analysis can be obtained from the ECMWF ERA5 reanalysis website: (https://www.ecmwf.int/en/forecasts/datasets/reanalysisdatasets/era5, European Centre for Medium-Range Weather Forecasts, 2022). TRACK is available for download from https://gitlab.act.reading.ac.uk/track/track.

## Appendix A

### A1   Further TRACK Details

In this work, TRACK is used to identify anticyclones corresponding to positive Z500 anomalies with respect to the instantaneous zonal mean component, once the climatological zonal mean anomaly is subtracted. Small scales are removed by spectral filtering, lowering the original resolution of the data to T42 resolution. Once the maxima in Z500 anomaly field are identified, tracks are constructed by finding nearest neighbours in consecutive time steps rather than the more sophisticated optimization method (Hodges, 1994, 1999) as there are typically only a small number of systems in any time step and blocks are often stationary features.

## A2 Sector Blocks Shorter than Five Days

There is an important distinction to be made between the persistence of the grid point $Z'_*$ blocking index, and the persistence of a sector blocking event. This is illustrated in the schematic shown in Fig A1. Each red box denotes a grid point that meets the "blocked" anomaly magnitude and persistence conditions via the $Z'_*$ index. In Days 1 and 2, a small group of grid points is blocked (area of $0.8 \times 10^6$ km$^2$), but not enough to exceed the sector blocking definition of $1.0 \times 10^6$ km$^2$. From Days 3–5, two additional grid points meet the 5-day persistence criterion, resulting in a group of grid points large enough such that sector blocking occurs. This sector block event only lasts for three days, as from Day 6 onwards the area of blocked grid points decreases to below the threshold again. This example shows how a slightly more mobile, or smaller, block may only meet the sector block threshold for a few days, despite the grid point level 5-day persistence criterion. A similar situation can arise when a larger block occurs outside of either the ATL or PAC domains, but the edge of the block is inside one of the domains. However in all of these scenarios, it is still possible that severe surface conditions can be brought about by these "edge cases", and indeed AC eddies can still help to form or maintain these blocks. Therefore, these "edge cases" are retained in the analysis.

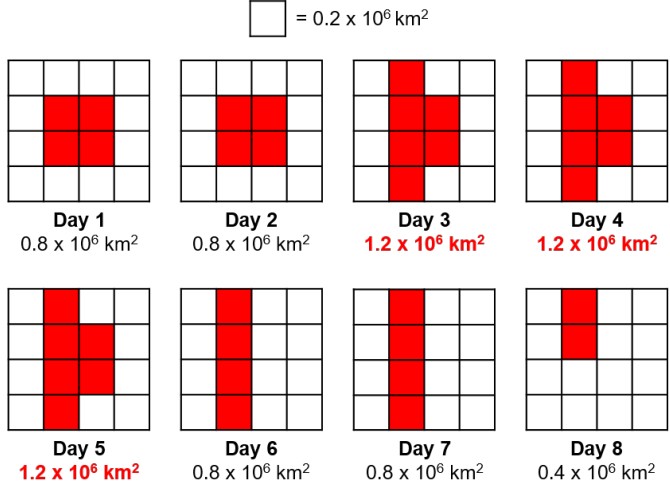

**Figure A1.** Schematic showing the difference between the persistence of the $Z'_*$ index at grid point level, and the persistence of a sector blocking event. For simplicity, each box represents an area of $0.2 \times 10^6$ km$^2$. Red boxes represent grid points that exceed the minimum $Z'_*$ threshold for blocking for a minimum of 5 days. The numbers below each day show the daily total blocked area in this scenario, with numbers in red showing days where the sector block area threshold ($1.0 \times 10^6$ km$^2$) is exceeded.

## A3 Block Persistence Distribution for MAM and SON

Histograms showing the block persistence distribution for MAM and SON in both the ATL and PAC regions is shown in Fig. A2. The distributions are broadly similar to those found in DJF for the respective regions, with similar Q1, Q2, and Q3 values.

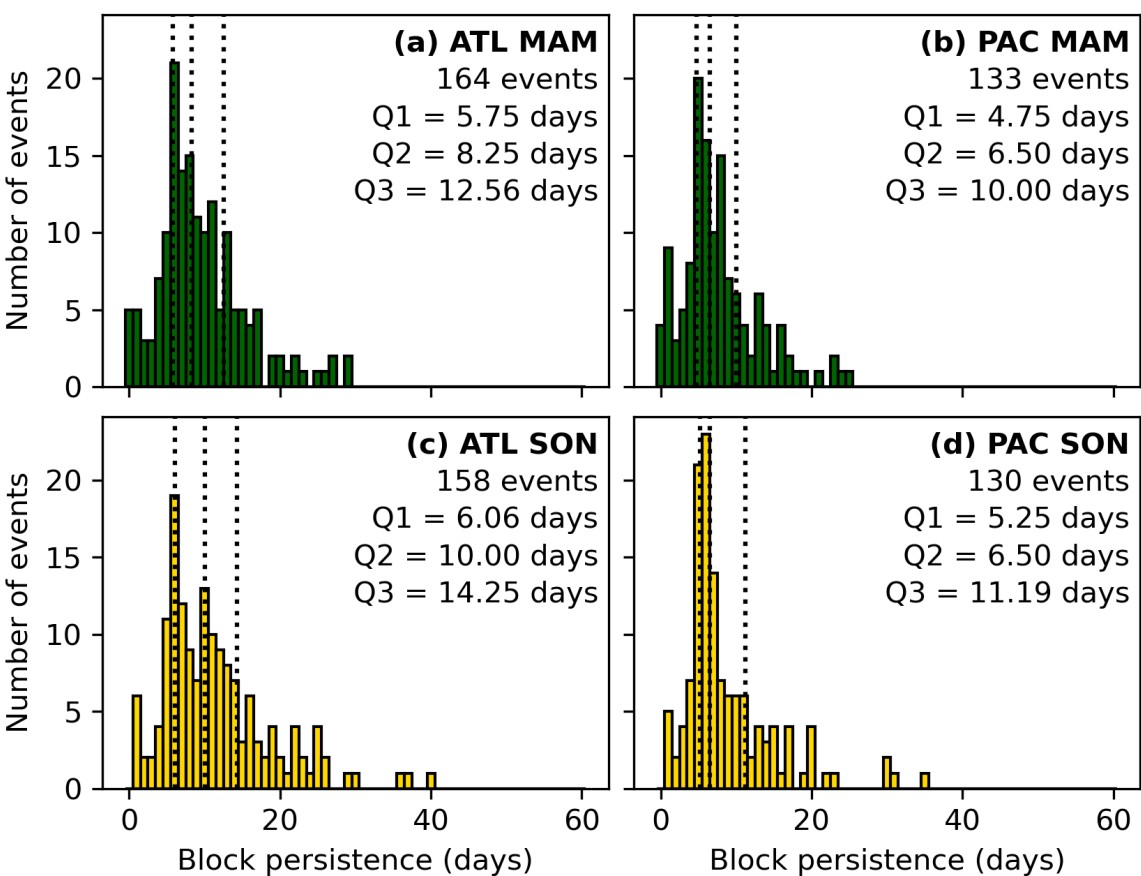

**Figure A2.** As in Fig. 6, but for MAM (a, c) and SON (b, d).

### A4   Number of AC Eddies Contributing to MAM and SON Blocks


The relationship between block area, persistence, and number of contributing AC eddies for MAM and SON for both the ATL and PAC domains is shown in Fig. A3. In both domains in both seasons here, the general patterns between the three variables is the same as those found in DJF. Pearson correlation coefficients are as high they are in DJF for these seasons between block persistence and the number of AC eddies (0.59–0.71), block persistence and area (0.62–0.73), and number of AC eddies and

block area (0.32–0.45).

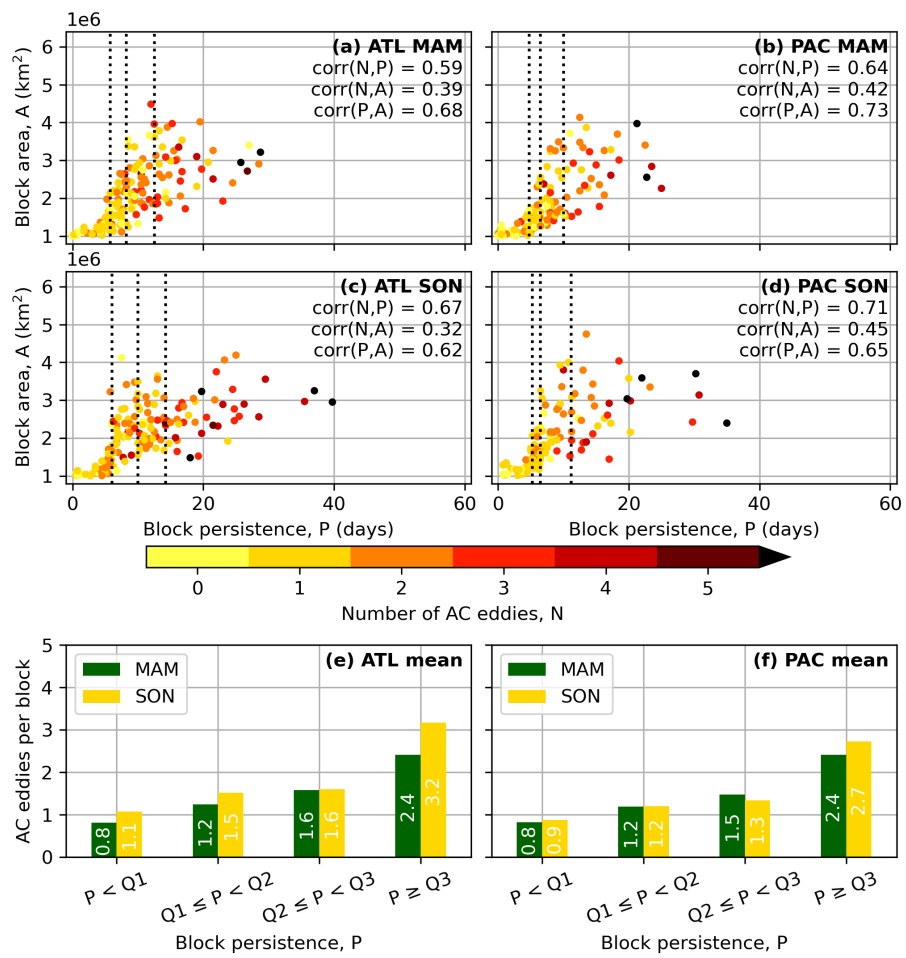

**Figure A3.** As Fig. 7, but for MAM (a, c), JJA (b, d), and ATL and PAC means (e, f).

## A5   AC Eddy Strength and Speed for Blocks in MAM and SON

The mean AC eddy intensity, zonal and meridional speeds for MAM and SON for blocks of all lengths are shown by the black lines in Figs. A4, A5. The eddies exhibit the same qualitative characteristics as those in DJF and JJA, but with intermediate values. Both speed and intensity for MAM and SON are also very similar to each other.

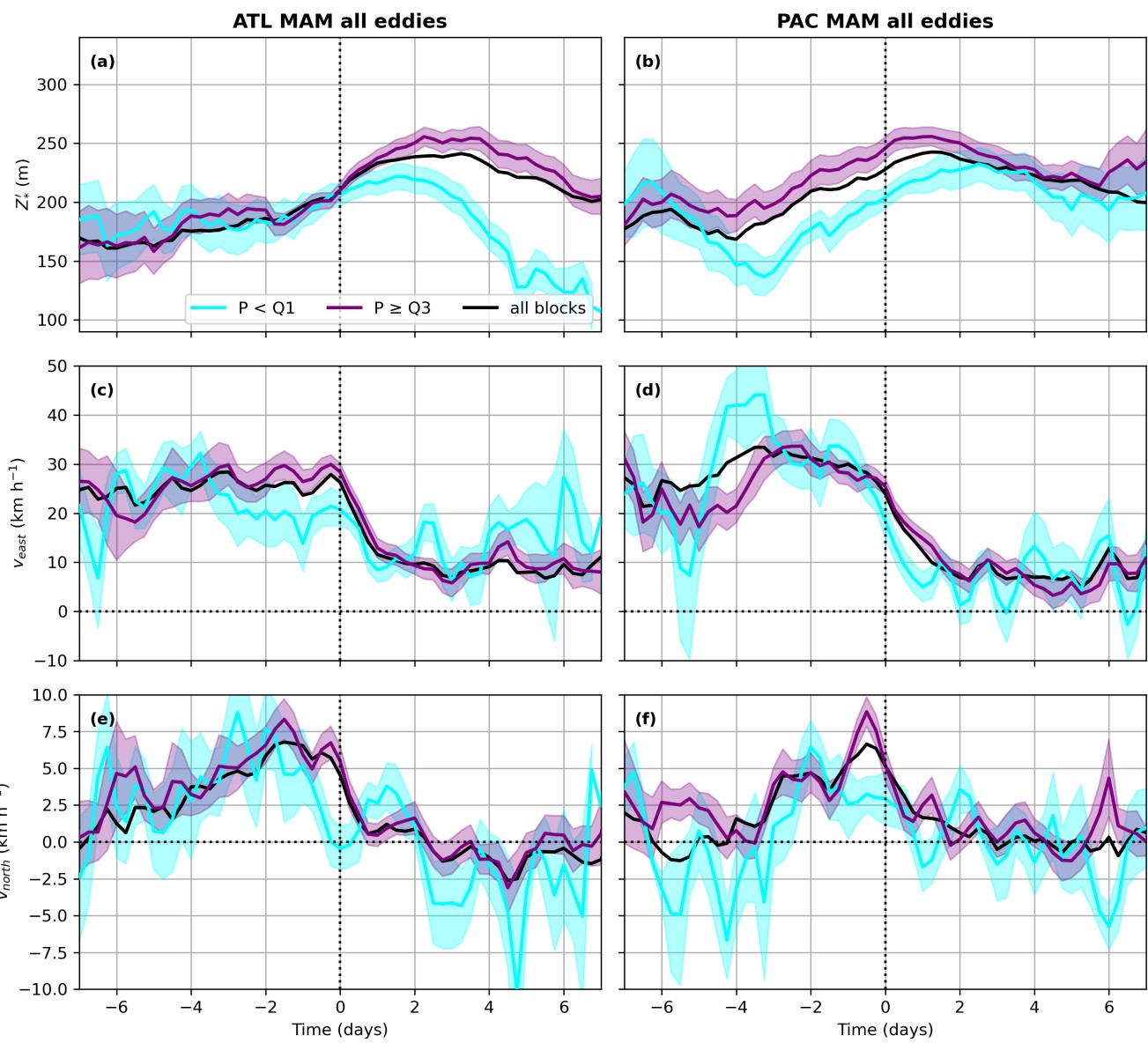

**Figure A4.** As in Fig. 8, but for MAM.

## A6 AC Eddy Strength and Speed for Blocks of Different Lengths in MAM and SON

AC eddy speed and strength for blocks of different lengths for MAM and SON is shown in Figs. A4 and A5. Generally, like in JJA and DJF, the strength and speed of the eddy is independent of the persistence of the block it contributes to. However,

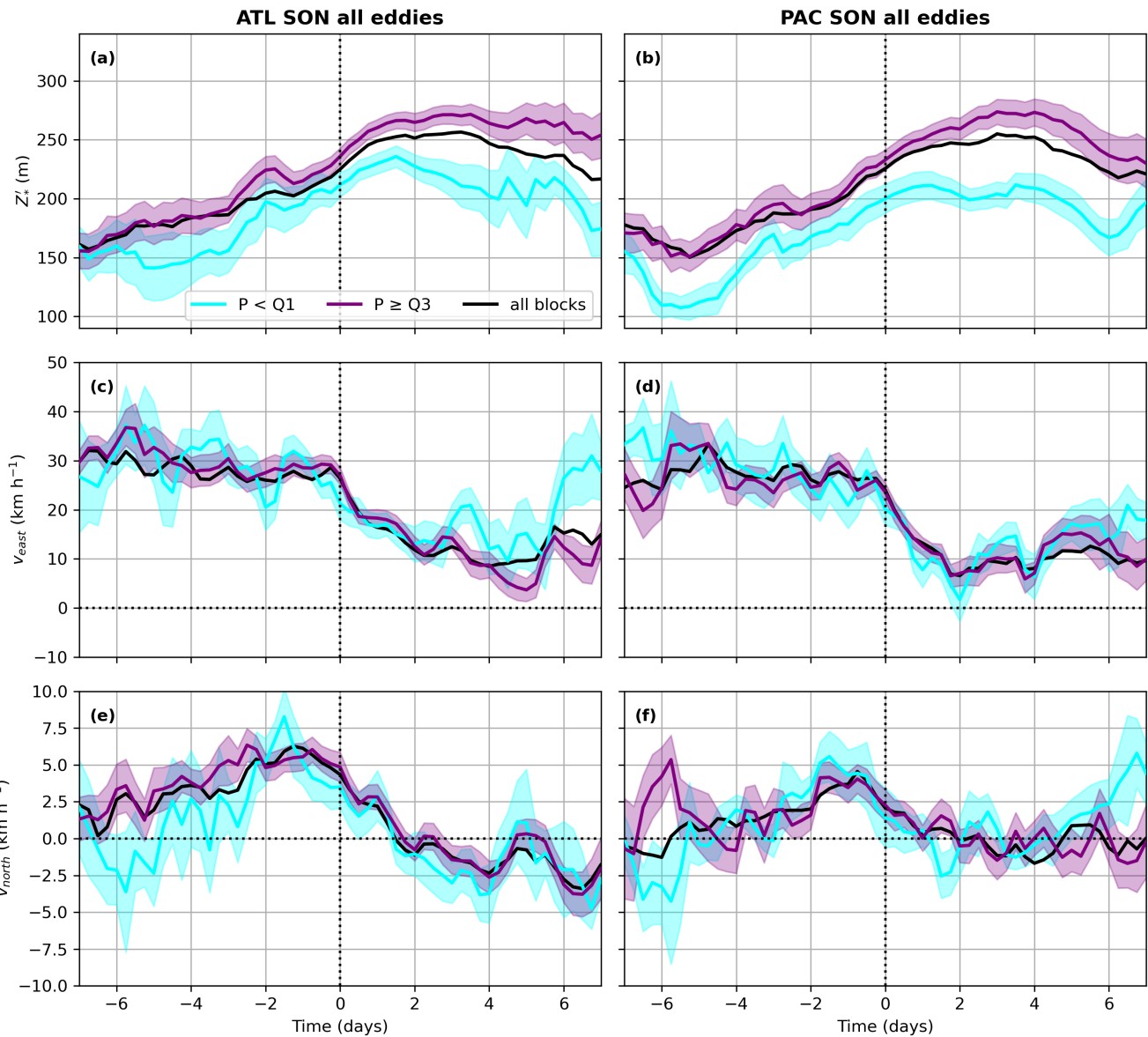

**Figure A5.** As in Fig. 8, but for SON.

MAM PAC eddies appear to also show some variation in their strength according to how long the block is, but the standard errors are large and sometimes overlapping.

*Author contributions.* CCS developed the blocking index, sector blocking definition, and the analysis of the results. OM-A, KIH, RKHS and DA secured PhD project funding for this work. KIH provided access to, and guidance with using TRACK. The manuscript was written by CCS, with support from OM-A, KIH, RKHS, and DA.

*Competing interests.* The authors declare that they have no conflict of interest.

*Acknowledgements.* We thank Dehai Luo and two anonymous reviewers for their constructive feedback which helped to improve the article. CCS is funded by the Natural Environment Research Council (NERC) via the SCENARIO Doctoral Training Partnership (Grant NE/S0077261/1) with additional CASE funding from the UK Met Office. OM-A, KIH and RKHS are supported by the U.K. National Centre for Atmospheric Science (NCAS) at the University of Reading (R8/H12/83/007). DA is supported by the Joint BEIS/Defra Met Office Hadley Centre Climate Programme (GA01101).

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
