# Peer review of "Transient Anticyclonic Eddies and Their Relationship to Atmospheric Block Persistence"

_Weather and Climate Dynamics, 2022_

## Community Comment (CC1)

Some comments on "Transient anticyclonic eddies and Their relationship to Atmospheric blocking persistence" by Suiters et al.

More recently, I have read a manuscript "Transient anticyclonic eddies and their relationship to atmospheric blocking persistence". I found that this manuscript is interesting, but this study is a phenomenological one, which cannot identify the causal relationship between transient anticyclonic eddies and blocking persistence. Because this study is in my research domain and the relation between blocking and transient synoptic eddies has been examined in my group before many years, below I give some comments on this manuscript to help the authors understand how transient eddies reinforce and maintain blocking and how intensified blocking deforms transient eddies.

**Comments:**

(1) In the daily geopotential height field of a blocking flow, the synoptic-scale anticyclonic (cyclonic) eddies are often seen to be intensified and shifted northward (southward) during the blocking growth and maintenance episodes (Berggren et al. 1949). Such an eddy deformation is also referred to as eddy straining or cyclonic wave breaking (CWB). Thus, many investigators inferred that eddy straining or CWB leads to blocking onset. Based on this, they also concluded that persistent eddy straining or persistent CWB leads to persistent blocking. In fact, the relationship between transient eddies and blocking is a chicken-egg problem. Thus, one cannot examine the causal relationship between the blocking persistence and transient anticyclonic eddies only using the identification method. However, this issue can be solved by a nonlinear theoretical model that considers a blocking as a nonlinear initial value issue of the blocking interacting with synoptic-scale eddies. The relationship between blocking and transient synoptic-scale eddies and what determines the persistence of atmospheric blocking has been widely examined in a nonlinear multi-scale interaction (NMI) model and has been clearly clarified (Luo

et al. 2005, 2014, 2019).

- (2) The mutual relationship between the blocking and synoptic-scale eddies has been examined in Luo (2005). It is found that the blocking and synoptic-scale eddies are dependent each other. Because of the feedback of the intensified blocking on the preexisting synoptic-scale eddies prior to entering the blocking region, the preexisting synoptic-scale eddies may slow down and undergo a north-south straining, which are dominated by deformed eddies with meridional tripoles.
- (3) Atmospheric blocking persistence has been first investigated by Yeh (1949), who found that atmospheric blocking tends to be long-lived in high latitudes. Luo et al. (2019) found from the NMI model that when the north-south gradient (PVy) of background potential vorticity is smaller, atmospheric blocking tends to be more persistent. Note that PVy is a modified β. When the background westerly wind or meridional temperature gradient is weaker or the latitude is higher, PVy is smaller. In this case, the blocking system has weaker energy dispersion and stronger nonlinearity so that the blocking can be more persistent. Thus, the persistence of atmospheric blocking is mainly determined by the background condition (i.e., the magnitude of PVy), rather than synoptic-scale eddies, even though synoptic-scale eddies have different phase speeds under different background conditions. As also noted by Luo et al (2019), the eddy forcing induced by preexisting synoptic-scale eddies prior to entering the initial blocking can be more persistent as PVy is smaller. In this case, atmospheric blocking can be more persistent due to persistent eddy forcing by preexisting synoptic eddies prior to entering the initial blocking.
- (4) The interaction between transient eddies and blocking satisfies the symbiotic relation noted by Cai and Mak (1990). The onset and intensification of atmospheric blocking not only depends on the spatial structure of preexisting synoptic-scale eddies prior to entering the initial blocking, but also the deformation of preexisting synoptic-scale eddies depends on the intensification of atmospheric blocking. Thus, the blocking and synoptic-scale eddies are coupled together and dependent each other. For an given initial blocking, the initial blocking can be amplified into a

typical blocking if preexisting synoptic-scale eddies ( $\psi'_1$ ) prior to entering the initial

blocking satisfy  $\frac{\partial q}{\partial t} \simeq -\nabla \cdot (\mathbf{v}'_1 q'_1)_p$  (Luo et al. 2014), where q is the PV anomaly of the initial blocking,  $\mathbf{v}'_1 = (-\partial \psi'_1 / \partial y, \partial \psi'_1 / \partial x)$  and  $q'_1 = \nabla^2 \psi'_1$ . In other words, when the eddy forcing  $-\nabla \cdot (\mathbf{v}'_1 q'_1)_P$  has the same spatial structure as the PV anomaly qof the initial blocking, a typical blocking can form from this initial blocking under the eddy forcing. When the initial blocking is intensified (Fig. 4a of Luo et al. 2014), the feedback of intensified blocking can cause the deformation of preexisting synoptic-scale eddies. In this case, deformed eddies ( $\psi_2'$ ) are produced. The daily synoptic-scale eddy field during the blocking episode can be represented by  $\psi' = \psi'_1 + \psi'_2$ . Because  $\psi'_2$  includes the amplitude of the intensified blocking, the synoptic-scale eddies in the daily synoptic-scale eddy field ( $\psi' = \psi'_1 + \psi'_2$ ) are inevitably intensified, split into two branches and slowed down with the growth of blocking (Fig. 4b of Luo et al. 2014). This case corresponds to eddy straining. In the daily total field (the sum of mean flow, blocking part and  $\psi' = \psi'_1 + \psi'_2$ ) of the blocking flow, anticyclonic (cyclonic) eddies are intensified and shifted northward with the blocking growth (Fig. A2), which corresponds to CWB (Fig. 4c of Luo et al. 2014). When the blocking is more persistent (Luo et al. 2019), more transient anticyclonic eddies are seen due to the persistent feedback of blocking because there is a symbiotic relationship between the blocking and anticyclonic eddies (Fig. 4c of Luo et al. 2014). This does not imply that persistent blocking is produced by more anticyclonic eddies.

(5) The authors concluded that blocks can be maintained through repeated absorption of anticyclonic eddies. In fact, blocking events do not always occur, but synopticscale anticyclonic eddies can often be seen. Why some anticyclonic eddies can be absorbed into the blocking, but others cannot. The authors should answer under what condition the anticyclonic eddies can be absorbed into the blocking to maintain it. This problem is easily explained in terms of  $\frac{\partial q}{\partial t} \simeq -\nabla \cdot (\mathbf{v}'_1 q'_1)_P$  because

only some of synoptic-scale eddies can meet this condition. When the preexisting synoptic-scale eddies ( $\psi'_1$ ) drive the onset and intensification of blocking (q), the feedback of blocking can cause the deformation of preexisting synoptic-scale eddies to result in the repeated absorption of anticyclonic (cyclonic) eddies by the blocking (Fig. A2 or Fig. 4c of Luo et al. 2014). When the blocking has longer lifetime, there are inevitably more anticyclonic (cyclonic) eddies within the blocking regions. Thus, persistent blocking can often occur together with more anticyclonic (cyclonic) eddies. But, this cannot lead us to conclude that more anticyclonic (cyclonic) eddies lead to persistent blocking.

I suggest that the authors should read the following references to improve the understanding of how the blocking and synoptic-scale eddies interact and what leads to the persistence of atmospheric blocking.

References:

- Berggren, R., Bolin, B. and C.-G., Rossby, 1949: An aerological study of zonal motion, its perturbations and break-down. *Tellus*, **1**, 14–37.
- Cai, M and M. Mak, 1990: Symbiotic relation between planetary and synoptic-Scale waves. J. Atmos. Sci., 47, 2953–2968
- Luo, D., 2005: A barotropic envelope Rossby soliton model for block-eddy interaction. Part III: Wavenumber conservation theorems for isolated blocks and deformed eddies,

J. Atmos. Sci., 62, 3839-3859

- Luo, D., J. Cha, L. Zhong, and A. Dai, 2014: A nonlinear multiscale interaction model for atmospheric blocking: The eddy-blocking matching mechanism. *Quart. J. Roy. Meteor. Soc.*, 140, 1785–1808, doi:10.1002/qj.2337.
- Luo, D., W. Zhang, L. Zhong and A. Dai, 2019: A nonlinear theory of atmospheric blocking: A potential vorticity gradient view. J. Atmos. Sci., 76, 2399-2427.
- Yeh, T. C., 1949: On energy dispersion in the atmosphere. J. Meteor., 6, 1-16.

Figure A1. Idealized sketches of the development of unstable waves at the 500 mb level, in association with the development of a blocking anticyclone in high latitudes. Cold air is in blue color and warm air in red. Solid lines are stream lines and broken lines the frontal boundaries (Taken from Berggren et al. 1949).

---

## Author Comment (AC1)

**Author's Response to Referee Comments on Suitters et al. (2022)**

We thank the two referees for their insightful comments in helping to improve the manuscript. We address their comments below. Reviewer comments are in black and our responses appear in blue. Figures included in our responses are labelled "R1, R2" etc. to avoid confusion with figures referenced in the paper.

**Referee 1**

This work focuses on the contribution of anticyclonic eddies to the maintenance of blocks, more particularly it investigates the relationship between anticyclonic eddies strength, zonal and meridional velocities, and the blocking persistence. Blocking events occurring in two areas of the Northern Hemisphere (the North Pacific and the North Atlantic/western Europe area) and during four seasons are studied here. The method used here to detect both the anticyclonic eddies and blocks is interesting. The science is sound; the article is well written and the figures and tables are clear. I have two major comments on this paper and a few minor comments.

Major comments:

- First, this paper is quite long for the number of results discussed here. I wonder if the authors could remove or shorten some sections:
  - Section 4.1 is a bit long as the result are very similar to previous studies. Figure 4 could be moved to the appendix and summarized in a couple of sentences.

We appreciate the fact that the results shown in Section 4.1 are similar to previous studies, but we feel that it is important to demonstrate that our blocking detection method behaves in a way consistent to previous studies. Since the $Z'_*$ index is introduced for the first time in this piece of work, we believe it is necessary that the climatology plots (Fig. 4) are included in the main text to highlight the similarities with other indices and therefore justify the usage of the index. Having said this, we do also appreciate that the description of the figure is a bit long and does indeed repeat many findings from previous work. Therefore, the second paragraph has been replaced by one sentence summarising the seasonality of the index. The discussion surrounding the magnitude of the blocking frequencies has been retained and altered slightly following a comment from Referee 2 (see below).

  - Section 4.2 could also be summarized and merged with section 5.1. In addition, the values shown in Tables 1, 2 and 3 could be directly added on the figures to shorten the paper.

We again feel this information is important to retain in the main body of the text. However, the discussion surrounding the figure in Section 4.2 (Fig. 5) has been made more concise and combined with the previous section where the spatial climatology was discussed. Sections 4.1 and 4.2 have therefore been combined (into a standalone Section 4) instead of combining 4.2 with 5.1 as suggested here. We think that Fig. 5 is a better fit with the spatial climatology discussion because this figure also focuses entirely on the blocks themselves, while Section 5 introduces the results concerning the AC eddies. The discussion of Fig. 5 has also been adapted to include a comment about the events lasting less than 5 days in JJA, reflecting the fact that a quarter of all sector blocks in JJA last less than 5 days. This indicates that the blocks are small in summer (so only marginally meet the area criterion for sector blocking), which we feel is an important climatological aspect of the blocks that cannot be obtained by simply looking at Fig. 4.

The comment about removing the tables and adding the information they contained was a very useful addition and we thank the reviewer for suggesting this. The number of events row from Table 1 and quartile definitions have been added to the panels in the histograms (Fig. 5), the correlations shown in Tables 2 and 3 have been added as text to the existing panels in the scatter plots (Fig. 6a-d), and the mean number of AC eddies rows in Tables 2 and 3 have been included as two additional panels (new bar charts in updated Fig. 6e-f). Language when describing "25th, 50th, 75th percentiles" in the text has been adapted to read "Q1/2/3", referring to the quartiles of persistence instead, for clarity and to be consistent with the new labels in the figures. Upon completing this task, three small errors were discovered and amended in the manuscript: (1) corr(N,A) in ATL JJA is in fact *__significant__* (not insignificant as previously asserted); (2) number of anticyclonic (AC) eddies per block have been correctly calculated now (the values in the table were incorrect previously); (3) Line 246 in updated paper: "Longer blocks are generally larger, and this relationship is strongest in *__summer__* in the PAC region" (previously stated "winter"). Removing the tables and providing this information in figure form was very helpful in finding these errors!

   o   The curves shown in Figure 7 could be directly added on Figures 8 and 9 to remove Figure 7. Also, Figure 9 could be move to the Appendix as it does not show strong differences between the shortest 25% and longest 25% of blocks.

We again found this particular comment especially helpful, not only in terms of data presentation but also helping with the interpretation of the results. We have updated the manuscript to remove Fig. 7 and amended Figs. 8 and 9. The updated Fig. 9 is shown below as an example (Fig. R1 here). We have implemented the suggestion of plotting the mean profile curve from Fig. 7 onto Figs. 8 and 9, and we have only plotted the profiles for the shortest 25% and longest 25% of blocks. The updated figures are much clearer and easier to interpret, with less clutter and using different colours to denote the properties of the eddies that contribute to differences in block persistences, instead of

[Figure]

**Figure R1.** Updated Fig. 9 (becomes Fig. 8 in the updated manuscript) showing the time series of mean strength **(a, b)**, zonal speed **(c, d)** and meridional speed **(e, f)** for AC eddies that contribute to ATL (left) and PAC (right) blocks. Only the eddies contributing to the shortest (cyan) and longest (purple) 25% of blocks are shown, along with the mean profile of AC eddies contributing to blocks of all lengths (black). Shading indicates standard error from the mean.

different line styles. Because we have combined Figs. 7 and 8/9, we have merged sections 5.2 and 5.3.

The updated figures have in fact made it easier to change a conclusion drawn from these profiles. The (main) conclusions for the ATL domain remain unchanged – eddy strength is only different for the shortest and longest 25% of blocks in DJF (and SON), in MAM and JJA there is no significant difference. However in the PAC domain, these cleaner plots allow for a slightly different interpretation to that described in the text. Eddy strength at the point of entering the block is significantly different between the longest and shortest 25% of blocks in **_all_** seasons: in DJF, MAM and SON, the most persistent blocks result from the interaction between stronger eddies than the shortest 25% of blocks, however in JJA the longest blocks actually interact with weaker eddies. With this behaviour now evident in the PAC sector, we feel that the plots for both DJF and JJA belong in the main body of the text, and not the Appendix as suggested here. Section 5.2 (was Section 5.3) has been rewritten to reflect these new conclusions, along with the appropriate changes in the Conclusions, Abstract, and Short Summary. Once again we thank the reviewer for this suggestion as it has led to an important change in the conclusions of our work!

- Second, the paper is quite descriptive and does not address the dynamics behind the relationship between anticyclonic eddies and blocks as could be thought after reading the introduction (e.g. lines 57-60 or lines 78-81) or the title. It does not show how these anticyclonic eddies dynamically contribute to the persistence of blocks. The space gained by summarizing some sections as suggested above could be used by the authors to develop more the dynamics behind this relation.

Thank you for pointing this out – we agree that the first version of our article was more statistical than at first thought given the introduction we provided. The space gained by implementing the changes you outlined above allowed us to add two short sections describing a little more the dynamics at play here: in the context of the case study (lines 195-208 in updated paper); and a "Discussion" section (new Sect. 6 in the updated article) in an attempt to describe a little more the dynamics behind our findings.

We approach the dynamical arguments by considering the effect of the AC eddies on both the area and intensity of the blocks. Yamazaki and Itoh (2013) argue that larger and more intense blocks are harder to advect downstream, since they require more forcing from upstream. We observe that often the block area and/or intensity increases upon the arrival of an AC eddy into the block. Therefore, we argue that AC eddy interactions increase the persistence of blocks by increasing their area or intensity.

The discussion section also addresses why the relationship between block persistence and AC eddies is slightly weaker in the ATL than the PAC. We argue that two conflicting scenarios are happening in the ATL domain, where synoptic-scale anomalies are more important over continental Europe than the Atlantic Ocean (Miller and Wang, 2022). Since our ATL domain covers both, we deduce that these factors may be contributing to a slightly weaker (but still relatively large) relationship than in the PAC.

We also address why it appears that JJA PAC blocks absorb weaker anomalies when they persist longer. This is again the result of us capturing, in essence, two different blocking centres here (one near the climatological blocking maximum to the west, and another one inside the domain itself). More details about all aspects mentioned above are provided in the new Discussion section.

Yamazaki, A. and Itoh, H., 2013. Vortex–vortex interactions for the maintenance of blocking. Part I: The selective absorption mechanism and a case study. Journal of the Atmospheric Sciences, 70(3), pp.725-742.

Miller, D.E. and Wang, Z., 2022. Northern Hemisphere winter blocking: differing onset mechanisms across regions. Journal of the Atmospheric Sciences, 79(5), pp.1291-1309.

Minor comments:

- Please, give more details in the Appendix on how the anticyclonic eddies are tracked.

The following paragraph has been added to the appendix:

In this work, TRACK is used to identify anticyclones corresponding to positive Z500 anomalies with respect to the instantaneous zonal mean component, once the climatological zonal mean anomaly is subtracted. Small scales are removed by spectral filtering, lowering the original resolution of the data to T42 resolution. Once the maxima in Z500 anomaly field are identified, tracks are constructed by finding nearest neighbours in consecutive time steps rather than the more sophisticated optimization method (Hodges, 1994, 1999) as there are typically only a small number of systems in any time step and blocks are often stationary features.

Hodges, K.I., 1994. A general method for tracking analysis and its application to meteorological data. Monthly Weather Review, 122(11), pp.2573-2586.

Hodges, K.I., 1999. Adaptive constraints for feature tracking. Monthly Weather Review, 127(6), pp.1362-1373.

- Line 112: Do you mean the monthly deviation from the zonal mean Z500?

Yes, this is what we meant. Thank you for pointing this out, it has been corrected in the text.

- Figure 3: Could you plot the continent lines in a distinct colour to separate it better from the geopotential anomaly contours. In addition, maybe you should plot only the "ongoing" tracks to make the figure cleared?

Thank you for the suggestion. In the updated plot, coastlines are now plotted in grey while the geopotential anomalies remain in black. We also agree that only having the "ongoing" tracks in each panel is clearer, so this change has also been implemented.

- Figure 5: why do you show the blocks lasting less than 5 days?

We have shown these events because they still meet our definition for "sector blocking". At the grid point level, a 5-day persistence criterion is indeed applied (as explained in Sect. 2.3), but when we define a "sector block" to be occurring, we require enough grid points to be meeting this persistence criterion inside the domain. If the system is small enough, it is possible that the area threshold is only met temporarily. A simplified example of this is shown in Fig. R2. Assuming the anomaly value of each red grid box exceeds 100m, each individual grid point in red satisfies the 5-day persistence criterion, so is classed as "blocked" according to the blocking index. However, when we invoke the $1.0 \times 10^6$ km$^2$ area threshold used to classify a blocking event, only in days 3-5 is there enough blocked points to exceed the threshold. Situations such as these occur fairly often in summer when blocks tend to be smaller (but may still be bringing impactful surface conditions), and are therefore kept in the histograms. A similar arrangement can happen when a block occurs predominantly outside of the domain, with only a small part of the block inside the domain (e.g. a Ural block with its

[Figure]

**Figure R2.** Schematic illustrating the difference between "grid point blocking" according to the blocking index, and "sector blocking". Each box represents an area of 0.2 x 10⁶ km². If a box is red, it satisfies both the 100 m anomaly threshold and 5-day persistence criterion imposed by the blocking index. A "sector block" occurs when an area of > 1.0 x 10⁶ km² is blocked in the sector (indicated by the red number below the plot for that day).

western flank clipping the ATL domain). Section 2.3 has been updated to include a short explanation of this scenario happening, and the discussion of Fig. 5 in the text has also been adapted to explain how these situations can occur fairly commonly in summer. Figure R2 has also been included in the Appendix for further clarification.

- Figure 5: Could you add in the figure the number of blocks in each area and season (as shown in Table 1)?

Thanks for the suggestion, the updated Fig. 5 has this information on it now.

- Lines 220-230: Could you add the values of the different percentiles in Figure 5?

This section has been rewritten as discussed in your earlier comment, and we have endeavoured to add the values of Q1, Q2, Q3 wherever we have stated them.

- Line 252-256: could you add the number of eddies in parenthesis as done in lines 250-251?

Once again this section has been slightly rewritten, but eddy numbers have been given in parentheses where appropriate.

- Figure 6: the colour of the dot plotted behind is not visible. Could the authors plot the relation between persistence and number of eddies in another panel? Or plot in another way the number of anticyclonic eddies

We appreciate the concern with this figure; however we feel this is the best way to display the data. If the data was plotted simply as number of eddies (N) on one axis with persistence (P) on another, there would be a much bigger issue with overlapping data points due to both values being discrete values. The area axis (A) was used in part because it allows the data points to be separated more easily, and provide a much "cleaner" plot. Having said this, we have reduced the size of the dots in the scatter plot which we believe has greatly improved the issue where certain points are not visible behind others, while appreciating that this is still not perfect. The additional panels (e) and (f) added to Figure 6 now also help in showing the relationship of larger N with larger P in a more concise way than the scatter plots do, without the issue of overlapping data points.

- Line 271: what is the duration of the anticyclonic eddies?

[Figure]

**Figure R3.** Duration of all AC eddies that contribute to the shortest (cyan) and longest (purple) 25% of blocks in the ATL (a, c) and PAC (b, d) domains in DJF (a, b) and JJA (c, d). Boxes show the interquartile range (IQR) of AC eddy duration at each stage of blocking, the red lines indicate the median, and the whiskers indicate 1.5 x the IQR. Outliers have been excluded from the plot. "Before" means times before the AC eddies first enter the blocks (left of the t = 0 line on Figs. 7-9 in the original text), "During" means times when the AC eddy coincides in space and time with a blocked grid point, and "After" means times after the AC eddy has first left the blocked grid points.

We thank the reviewer for this interesting question. A boxplot showing the duration of the AC eddies before, during and after blocking (and total lifetime) for DJF and JJA for the longest and shortest 25% of blocks is shown in Figure R3. The main findings from these plots are explained briefly here. Firstly, for both long and short blocks in both DJF and JJA, AC eddies have a much shorter duration after blocking than they do before/during blocking. And for the longest 25% of blocks, over half of the AC eddies contributing to their maintenance do not have any life after the block (median = 0 days); i.e. the majority of AC eddies contributing to the longest blocks decay in, or become fully absorbed by, the block itself. Meanwhile, apart from PAC JJA, the majority of AC eddies contributing to the shortest blocks do have a lifetime after blocking (median > 0 days). These findings are perhaps not surprising and match with the line plots in updated Figs. 7 and 8 in that after the eddies have entered the block (t > 0), those contributing to the shortest 25% of blocks are faster-moving (and therefore no longer quasi-stationary, which blocking requires) than those contributing to the longest 25%.

Second, an unsurprising related result confirmed by the boxplots in Fig. R3 is that the AC eddies contributing to the longest blocks have a longer residence time inside the block than those associated with shorter blocks. The increased persistence of these blocks is therefore related to the increased duration and quasi-stationarity of the AC eddies inside the block (again c.f. updated Figs. 7 and 8 in the text).

Another finding is that AC eddies that contribute to blocks (of all lengths) have a longer total duration in JJA than in DJF. Once again, by also looking at the $v_{east}$ curves in the updated Figs. 7 and 8, we see that this longer lifespan is associated with slower zonal progression in JJA than DJF. These arguments are consistent with the fact that the summer jet is weaker and generally more variable, thus meaning that AC eddies progress more slowly along it than in winter. The ~25-50% increase in

duration (in the median) in JJA from DJF marries with a ~25-50% reduction in zonal speed from DJF to JJA (updated Figs. 7, 8), implying that a similar total distance is travelled by the eddies in both seasons (excluding the few very long eddy tracks shown in Fig. R6 in response to Reviewer 2).

Finally, we find that even for the most persistent blocks, the AC eddies that contribute to them, for the most part, do not stay in the block for longer than 5 days, especially in winter. This means that in order for a block to persist for much longer than this, repeated AC eddies may be required to "top up" the anticyclonic anomaly inside the block in order for it to persist. This finding was displayed in the scatter plots (Fig. 6) and was the main conclusion of this paper.

All things considered, we do not believe that the addition of this figure would add any new information to the manuscript, since all points mentioned above can already be explained by existing figures. However, these findings are summarised and a few additional comments have been added to the main text of the paper explaining the duration of the eddies, where appropriate.

- In Figures 7, 8 and 9, there is no separation between the through and absorbed eddies?

The reviewer is correct here in that the profiles plotted on the original Figs 7-9 considered all AC eddies and did not distinguish between those that are absorbed (finish inside the block) and those that pass through the block. A brief comparison between these two types of eddies is provided here.

The profiles of absorbed AC eddy intensity, zonal and meridional speed for all blocks, the shortest 25%, and the longest 25% of DJF blocks are shown in Fig. R4. The profiles for absorbed eddies are qualitatively similar to those shown for all AC eddies for strength, zonal and meridional speed. (Note that the mean line stops just after day+6 for the shortest 25% of blocks – this is expected since these eddies are now absorbed by the block, but these blocks are short so do not last longer than 6 days). The JJA profiles for absorbed eddies are also qualitatively identical to those for all eddies.

For the most part, the through AC eddies also behave the same as all AC eddies do. The only exception is that for ATL DJF blocks, there is no discernible difference in the strength of the absorbed

[Figure]

**Figure R4.** As in updated Fig. 7 in the manuscript, but only showing the characteristics of the absorbed AC eddies.

eddies at t = 0 for the longest and shortest 25% of blocks (not shown here). Otherwise, the same conclusions can be drawn as those for absorbed and all AC eddies.

Therefore, we decide not to separate the eddy profiles shown in updated Figs. 7 and 8 into those that are "absorbed" and those that go "through" the block. We have however added a sentence in Section 5.2 confirming the similarities between absorbed and through eddies.

- Figures 8 and 9: could you plot the shortest 50% of blocks and the longest 50% in a distinct colour to better differentiate the standard error.

This has been addressed as explained above. Instead, we plot results only for the shortest and longest 25% of blocks as this is where the starkest differences are. These are now plotted in different colours.

- Line 335: replace "Selective Absorption Mechanism" by "SAM".

Thank you for pointing this out, this has been amended in the text.

**Referee 2**

Review of "Transient Anticyclonic Eddies and Their Relationship to Atmospheric Block Persistence" by Charlie C. Suitters, and coauthors

The authors investigate the relationship between block persistence and synoptic-eddy (especially anticyclonic anticyclone) characteristics based on the traditional eddy-feedback mechanism originally proposed by the famous paper Shutts (1983). The authors applied a cyclone-tracking method to the synoptic anticyclones that interact withã€blocking, and discovered that i) longer blocks interact with more anticyclonic transients than less persistent blocks and ii) there is little relationship between the strength of the anticyclonic eddy and the blocking longevity except winter. In the manuscript, the authors comprehensively reviewed the blocking maintenance mechanism based on the eddy feedback mechanism and the results obtained here support importance of the eddy feedback mechanism. Also, the authors quantify the eddy feedback mechanism from both the Eulerian and Lagrangian perspectives. This paper include a lot of novel topics on the blocking mechanism and can develop the traditional eddy feedback mechanism from 1980s.

The reviewer evaluates that the manuscript is suitable for the journal Weather and Climate Dynamics that has the scopes on midlatitude dynamics, in which blocking and synoptic eddies are essential, though I also have a major comment about the correspondence between the anticyclone tracking used in this study and the Lagrangian tracking ways commonly used in previous studies. Then, the reviewer would suggest the paper is in a category of major revision. Specific comments are below.

Major comments

The authors define the anticyclonic eddies as positive anomaly from the zonal and temporal means, which seems different from typical cyclone tracking and particle tracking methods used in many studies. I think although the Lagrangian tracking used in this study is valuable to understand the eddy characteristics, but also think that many Lagrangian tracking schemes focus on the absolute (raw) fields (values) rather than their anomaly fields. Yamazaki and Itoh (2013a) mention in their paper that (raw) low PV supply is important for the blocking maintenance, and their Lagrangian tracking was done by raw (unfiltered) wind fields. More recent papers by Pfahl et al. (2015) or Yamamoto et al. (2021) which adopted the Lagrangian analysis into the blocking formation and maintenance mechanisms monitored (raw) PV values of tracked air parcels. Here, my question is that if you define the eddy intensity (strength) as a raw value (say, PV) in Figs. 7-9, does your conclusion that "there is a less clear relationship between block persistence and the strength of the AC eddies that it absorbs" change? For example, could you trace the mean column-averaged value of raw PV of 150-500 hPa (Schwierz et al. 2004) at or within an AC eddy? In addition, I think that Z anomaly as the eddy strength can be changed if latitudinal position of the eddy varies. In such perspective, I am wondering how are the track distributions of ACs that interact with blocking? To check the distributions and/or the statistics of the AC tracks may be helpful on your conclusion. The results by Yamamoto et al. (2021) might be useful.

We thank this reviewer for a thorough comment on our work. We have split our answer to these into four parts:

Could we trace the mean column-averaged value of raw PV of 150-500 hPa at or within an AC eddy?

We appreciate that previous studies have used this measure of PV to define AC eddies (e.g. Schweirz et al., 2004; Hauser et al., 2022) and blocking. However Z500 is an equally valid measure to define both metrics, and indeed has already been used for AC anomaly tracking (see Liu et al., 2018). A Z500-based blocking index and AC eddy definition was pursued as this is a more readily available diagnostic in reanalysis and model datasets, so can more easily be implemented elsewhere, if desired.

A PV-based method would be expected to give similar, but admittedly not identical, results to those presented here. Generally speaking, one would expect that ridges in Z500 would align with PV-derived ridges (through the PV invertibility principle). Generally speaking, the Z500 field is also spatially smoother than the full PV field, and is therefore easier to track anomalies in. The use of a Z500-based metric in our study also allows for theoretical block maintenance mechanisms explained using PV thinking, like the SAM, to be "tested" using another variable. Yamazaki and Itoh (2013) explain that longer-lasting blocks are larger and more intense than smaller ones, and we have shown (in the new "Discussion" section in the paper, Sect. 5) that Z500-based anomalies increase block area and/or intensity. Combined with our finding that more persistent blocks result from more (and sometimes stronger) eddies, we can deduce that our method is appropriately capturing the PV-based behaviour that the SAM describes, so is therefore valid to determine AC eddies with.

Having said this, current/future work by the authors of this manuscript is looking to attribute PV to AC eddies. In particular, the PV budget of the AC eddies at different stages of blocking is being looked in to, for a few selected case studies. Initial analysis on one case study period has shown that the AC eddy track centres used in this study align very well with local minima in vertically-averaged 150-500 hPa PV. An example of this is shown in Fig. R5. Therefore, we would expect a PV-based method to produce similar results to those presented here with our Z500-based method.

[Figure]

[Figure]

[Figure]

**Figure R5.** Vertically-averaged (500-150 hPa) PV (VAPV, shading), AC eddy track derived using the Z500-based method as described in Suitters et al. (2022) (white line, position at valid time in yellow dot), and block outline, again using the Z500-based technique (red). The dynamic tropopause (PV = 2 pvu line) is shown in black

Liu, P., Zhu, Y., Zhang, Q., Gottschalck, J., Zhang, M., Melhauser, C., Li, W., Guan, H., Zhou, X., Hou, D. and Peña, M., 2018. Climatology of tracked persistent maxima of 500-hPa geopotential height. Climate Dynamics, 51, pp.701-717.

Hauser, S., Teubler, F., Riemer, M., Knippertz, P. and Grams, C.M., 2022. Towards a diagnostic framework unifying different perspectives on blocking dynamics: insight into a major blocking in the North Atlantic-European region. Weather and Climate Dynamics Discussions, pp.1-36.

Schwierz, C., Croci-Maspoli, M. and Davies, H.C., 2004. Perspicacious indicators of atmospheric blocking. Geophysical research letters, 31(6).

Does the conclusion that there is a "less clear relationship between block persistence and the strength of the AC eddies it absorb" change if we define eddy intensity using raw PV?

Firstly, it should be noted that thanks to a comment by Reviewer 1 (above), we have slightly changed our conclusions surrounding block persistence and AC eddy strength. Now, we conclude that in the PAC region, AC eddy strength does have a relationship with block persistence, and in the ATL the same is true for DJF and SON. So, we are now able to conclude that more often than not, there is an important relationship between AC eddy strength and block persistence: (1) longer blocks result from stronger AC eddies in ATL DJF, ATL SON, PAC DJF, PAC MAM, PAC SON; (2) longer blocks result from weaker AC eddies in PAC JJA. Only in ATL MAM and ATL JJA is there no significant difference in AC eddy strength for the longest and shortest blocks. The text has been amended to reflect this new conclusion.

As discussed above, we expect that most of the time, the Z500-based AC eddies would align fairly well with the equivalent PV-based ones. We would therefore expect a similar relationship to hold true between AC eddies and block persistence if we instead defined them using raw PV. As mentioned above, certain blocking case studies will be tackled from a PV perspective in future work by these authors, so this hypothesis can be tested there instead. We do not think that this current study calls for PV thinking.

Can the Z anomaly (as a metric for eddy strength) be changed if latitudinal position of the eddy varies?

We have tried to normalise the eddy strength by its location as much as possible in our anticyclonic anomaly calculations. The strength of the eddies at any time depends on two things:

(1) The eddy's instantaneous Z500 anomaly from the zonal mean, $Z_*(\lambda, \phi)$. This value depends on the eddy's location (both its longitude $\lambda$, and latitude $\phi$) and its magnitude depends on (a) how amplified the eddy's ridge is and (b) how amplified the rest of the wave guide is at that longitude. If the eddy occurs at a time where the rest of the jet is fairly zonal at that latitude, a move northwards would indeed mean an increase in eddy strength. However, an eddy that occurs among an amplified jet, with many ridges and troughs, would not necessarily strengthen as it moves north. However it should be noted that blocking is associated with the poleward transport of higher-than-normal geopotential heights, so an increase in eddy strength as the eddy moves northward is not unexpected and perhaps even desirable behaviour.

(2) The background climatological wave pattern, $\overline{Z_*}(\lambda, \phi)$, quantified by the monthly climatological anomaly from the zonal mean at a grid point, as shown in Fig. 1 in the article. This has the effect of "normalising" the anomaly strength by location and time of year, meaning that any strengthening of the eddy could be considered "real", rather than just an artefact of the method.

What are the track distributions of the ACs that interact with blocking?

Thank you for this interesting comment. We show the AC eddy tracks, genesis and lysis locations for all ATL in DJF and JJA in Fig. R6. In both summer and winter, the overwhelming majority of AC eddies that contribute to ATL blocking travel from west to east. Most AC eddies contributing to ATL blocking begin in the North Atlantic storm track region before travelling along the wave guide and entering the blocking region. This result agrees well with the finding from Ioannidou and Yau (2008) concerning where anticyclonic Z500 anomalies are formed (their Figs. 6a, 8a). Some AC eddies are generated over continental North America and even the North Pacific, and this is more common in DJF than JJA. Most blocking AC eddies finish in the ATL region itself, though a sizeable proportion propagate further downstream once contributing to blocking and decay over central/eastern Eurasia (these are the "through" eddies), again this being most common in DJF. Panels (a) and (c) confirm the results indicated in response to Reviewer 1 above, in that summer eddies travel less distance than winter ones (Fig. 3 in this document). A similar pattern is evident for PAC blocks (not shown).

[Figure]

**Figure R6.** AC eddy track distributions for ATL blocks in winter (a, b) and summer (c, d). All AC eddy tracks are shown in the left column, and AC eddy track genesis (dark red) and lysis (pale orange) locations are shown in the right column. The ATL domain is shown in green.

Ioannidou, L. and Yau, M.K., 2008. A climatology of the Northern Hemisphere winter anticyclones. Journal of Geophysical Research: Atmospheres, 113(D8).

Yamamoto, A., et al., 2021: Oceanic moisture sources contributing to wintertime Euro-Atlantic blocking, Weather Clim. Dynam., https://doi.org/10.5194/wcd-2-819-2021.

Hauser, S., Teubler, F., Riemer, M., Knippertz, P. and Grams, C.M., 2022. Towards a diagnostic framework unifying different perspectives on blocking dynamics: insight into a major blocking in the North Atlantic-European region. Weather and Climate Dynamics Discussions, pp.1-36.

Other specific comments

1. Related to the major comment, could you show the trajectory statistics of synoptic cyclones? Since sometimes a Berggren-type blocking where there are several isolated anticyclonic or cyclonic vortices within the blocking region (e.g., Luo 2005) can exist.

While we agree that showing the trajectories of synoptic cyclones would be interesting, and often an important part of blocking, we argue that this is out of the scope of this work. The main aim of this study is to examine the extent to which synoptic-scale *anticyclones* extend the lifetime of blocking *anticyclones*. We decided to focus on the anticyclonic aspect of blocking for a couple of reasons. Firstly, the majority of previous studies have developed block detection methods purely in terms of how anticyclonic the region is, so our method is consistent with prior work. Secondly, it should be noted that not all blocking patterns are necessarily associated with quasi-stationary cyclones, but all blocks do have quasi-stationary anticyclones by definition (see Fig. 1 in Woollings et al., 2018).

We do however appreciate that tracking the cyclonic eddies associated with blocking is an interesting area of research. We have added a comment in the conclusions section about this

interesting avenue of discussion. We point the reviewer to e.g., the work of Maddison et al. (2019), who looked at the relationship between upstream cyclone activity and block predictability, since cyclones are out of scope for this work.

Woollings, T., Barriopedro, D., Methven, J., Son, S.W., Martius, O., Harvey, B., Sillmann, J., Lupo, A.R. and Seneviratne, S., 2018. Blocking and its response to climate change. Current climate change reports, 4, pp.287-300.

2. How do you obtain "u" and "v" in Figs. 7 and 8? If those values are Eulerian-based (raw) winds which are the interpolated wind values at the AC centers from the ERA5 gridded data, to what extent are those values different from the Lagrangian speeds of ACs obtained by your tracking method?

Thank you for pointing this out, as it has led to the need for a clarification in our text. When we mention "u" and "v", we were simply describing the zonal and meridional propagation speeds, respectively, of the AC eddies (i.e. the Lagrangian speeds obtained by TRACK). We have changed the notation in the text and figures to $v_{east}$ and $v_{north}$ instead to distinguish them from the commonly-used notation for wind components.

3. Several previous papers may be useful for the introduction part:

- Zhu et al. (2007) investigated the statistics between the synoptic cyclone activity and the Aleutian low intensity.

We thank the reviewer for pointing out this study, however we feel that it is not necessary to point this out in the introduction. Zhu et al. (2007) do not examine how synoptic cyclone activity effects the *persistence* of the Aleutian low, whereas we are explicitly exploring the relationship between synoptic anticyclone activity and block *persistence.* We therefore do not feel that the introduction needs to reference this paper.

- Okajima et al. (2021) proposed a new detection method for anticyclonic and cyclonic eddies based on curvature.

- Shi and Nakamura (2021) proposed a blocking detection index based on the Rossby wave breaking.

Thank you for pointing us towards these papers. These references have been added to the discussion in the Introduction regarding blocking indices/feature tracking.

Minor comments:

- L193-194: Why are the climatological frequencies different (16% vs 30%)?

The two climatologies have different frequencies because in our work, we consider the background anomalies from the zonal mean, $\overline{Z_*}$, leading us to achieve a smaller (and probably more realistic) block frequency than that used in Liu et al. (2018). Without removing the climatological wave pattern, our blocking index would be far too sensitive in winter, falsely detecting "climatological" conditions as being blocked in Western Europe, for example, because of the naturally high $Z_*$ here. Therefore, by removing the climatological anomaly from the zonal mean, we are able to identify actual blocks in this sector while reducing the number of false positives drastically.

Liu, P., Zhu, Y., Zhang, Q., Gottschalck, J., Zhang, M., Melhauser, C., Li, W., Guan, H., Zhou, X., Hou, D. and Peña, M., 2018. Climatology of tracked persistent maxima of 500-hPa geopotential height. Climate Dynamics, 51, pp.701-717.

- The term "standard error": Is it "standard deviation"?

We mean "standard error *from the mean*" when we mention "standard error". However, standard error from the mean is simply the standard deviation normalised by the square root of the number of eddies at each time point. In this way, we have also considered the sample size of eddy tracks at each time step – the measure of uncertainty is larger where we have fewer tracks. We have clarified the text to explicitly state "standard error ***from the mean***" to hopefully clear up any terminology confusion.

- L325 and L335: the abbreviation "SAM" is used before "Selective Absorption Mechanism"

Thank you, this has been amended in the text.

Refereces:

- Luo, D., 2005: A Barotropic Envelope Rossby Soliton Model for Block–Eddy Interaction. Part I: Effect of Topography, J. Atmos. Sci., https://doi.org/10.1175/1186.1.

- Okajima, S., et al, 2021: Cyclonic and anticyclonic contributions to atmospheric energetics, Sci. Rep., https://doi.org/10.1038/s41598-021-92548-7.

- Shi, N., and H. Nakamura, 2021: A New Detection Scheme of Wave-Breaking Events with Blocking Flow Configurations, J. Clim., DOI: 10.1175/JCLI-D-20-0037.1.

- Yamamoto, A., et al., 2021: Oceanic moisture sources contributing to wintertime Euro-Atlantic blocking, Weather Clim. Dynam., https://doi.org/10.5194/wcd-2-819-2021.

- Zhu. X., et al., 2007: A Synoptic Analysis of the Interannual Variability of Winter Cyclone Activity in the Aleutian Low Region, J. Clim., DOI: 10.1175/JCLI4077.1.

---

## Author Response (AR2)

**Responses to Second Round of Reviewer Feedback, and Community Comment by Dehai Luo, for Suitters et al. (2022)**

Here, we respond to the second round of reviewer feedback. We also respond to the community comment posted by Dehai Luo (page 8 onwards). Our responses are in blue text, with figures labelled by Fig. AR1, AR2, etc. ("Author Response").

**Reviewer #1**

Main comment:
- Lines 349-350: Why not defining sectors that change with seasons to be better adapted to the blocking main occurrence areas (like, for example, in Davini and D'Andrea, 2020)?

Thank you for this comment – this is something we did consider earlier on in the process of completing this study. We chose to define our sectors in such a way as to combine large blocking frequencies and the potential for large-scale impacts. Understanding the dynamics of blocking in areas where blocks are more likely to cause substantial hazards is arguably more useful than doing the same analysis where blocks are less impactful (though of course, our method could be applied anywhere). With these considerations in mind, we settled on the ATL and PAC sectors as shown in Fig. 2 in the paper. In the ATL sector, the climatological blocking frequency maximum in the Euro-Atlantic region is within, or very near to, the ATL sector for all four seasons. Therefore, results here are largely independent on the exact position of the ATL domain.

We concede that the same cannot quite be said for blocking in the North Pacific-North American region. The climatological blocking maximum is comfortably inside the PAC domain in DJF, MAM and SON (c.f. Fig. 2). Thus, moving the domain in these three seasons would, like in the ATL, have little bearing on the main conclusions of our work. However, the positioning of our PAC domain means that in summer, some of the highest block frequency contour is outside our region of analysis. The PAC domain was therefore designed to be a compromise between getting as much of the maximum

[Figure]

**Figure AR1.** *As in Fig. 3 in the main text, but for the extreme Canadian heatwave in June 2021. The PAC domain, as used in our study, is shown by the green box; shifting the domain westwards to align with the JJA climatological maximum block frequency would result in this event not being considered in our analysis.*

blocking frequency all year-round, while also ensuring that the blocks we analyse have the potential to bring impacts to populous areas. If the PAC domain was shifted westwards in JJA to overlap with the whole blocking frequency maximum at this time of year, the blocks we would have investigated would have been less impactful since they would have mostly been over the ocean. However, keeping the PAC domain fixed for all four seasons still allows us to include impactful blocking events in this region, even in summer, for example the extreme Canadian heatwave in June 2021 (see Fig. AR1). A PAC domain more aligned with the seasonal JJA maximum would have not captured the dynamics of this blocking event. See also the response to the main comment from Reviewer #2 below for a more in-depth analysis of the effects of moving the PAC and ATL domains.

In addition to the explanation already present in Sect. 2.3 in the text outlining our choice for the PAC and ATL domains, an additional sentence has been added summarising the above: "For example, the PAC domain as defined in this study is able to capture the dynamics of the severe North American heatwave in June 2021, which would not be the case if the domain was positioned closer towards the climatological summer blocking frequency maximum."

Minor comments:
- Lines 196-198: I think that the authors should replace the word "block" by "sector block" to avoid any confusion with the Scandinavian blocking present to the east of the area. Indeed, the area blocked and block intensity are computed only within the ATL sector. These metrics do not take into account the grid points that are part of the Scandinavian blocking but that are outside the ATL area.

Thank you for this suggestion. Minor changes have been made in this paragraph (changing "block" to "sector block") to distinguish between the blocked points in the ATL domain, and the blocked points outside the domain (Scandinavia) like you point out.

- Line 202: It should say "The arrival of the first AC into the sector is associated with the sharp increase in sector block area…". Indeed, on 27/02/2011, the AC eddy and the Scandinavian block had not merged yet as shown in Fig. 3c.

Thank you – this has been amended in the text.

- Line 274-275: except for DJF PAC sector blocks.

Thank you for pointing this out. The text has been amended account for this (the text already mentions this further on, so this sentence has been clarified).

- Line 277-278: the strengthening starts one day before entering sector block.

This is correct. An additional clause has been added to this sentence stating that some strengthening also occurs one day before entering the block, like you suggest.

- Line 320: What does "here" refer to? I find this paragraph a bit confusing as it starts with the weakening and slowing of the eddies and finishes talking about change in eddy intensity when they enter the sector block.

We appreciate that this sentence was not clear, and the paragraph was talking about both eddy speed and intensity, which was confusing to the reader. The sentence containing "here" has been removed, and this paragraph has been split into two to make a clear distinction between the discussion about the weakening and slowing, and another separate paragraph about the intensity.

- Lines 336-337: Within the framework of this theory, how do the authors explain that some AC eddies pass through the block or are spawned by the block?

"Spawned" and "through" eddies are interesting but have not been given much attention in the paper, since we are only interested in the interactions of AC eddies in terms of block persistence. To our knowledge, we are not aware of any work where AC eddies emerge from a block and propagate downstream. We have seen a few cases in our dataset where these "spawned" or "through" AC eddies then go on to interact with another block downstream, so these eddies certainly require further investigation – a comment on this has been added to the Conclusions section:
"…Furthermore, our results have also highlighted the existence of two further types of AC eddies, namely those that pass through the block, and those that are spawned by the block and propagate downstream. These types of AC eddies require further investigation, particularly as it is possible that they can go on to interact with another block event downstream…". In terms of explaining them theoretically, we hypothesise that these types of eddies are the result of eddy straining, when incoming eddies split into a northward and poleward branch. This splitting of eddies inside of blocks could result in only one eddy vortex merging with the main block, with the other component continuing to propagate downstream. Investigation of these eddies is out of scope for this study but as mentioned above, would be interesting to look at in future.

- Line 340: This sentence is a bit too assertive. In winter and autumn, there is a difference in the intensity of the AC in the ATL sector between the shortest and longest blocks

Thank you for pointing this out. We meant this in the sense that AC eddy intensity only has an impact on block persistence in some seasons in the ATL, not in all times of year like in the PAC sector. This has been clarified in the text.

- Line 383: It is not that clear: in DJF in ATL sector, eddies intensify after entering the 25% shortest blocks. It is also true in JJA in PAC sector.

We have added the word "generally" to this sentence to suggest that this is not always the case, as you rightly point out.

Typos:
- Line 108: "produce" instead of "proudce"
- Line 246: correlation of 0.71
- Line 374: "... relationship between block persistence...". Between is missing.
- Line 381: citation in parenthesis
- Line 382: remove "is"

Thank you for kindly pointing out these typos, we have corrected them in the text.

**Reviewer #2**

Comment:
After reading your revised manuscript in L426-428 ("When the PAC domain is shifted ... (Fig. 9a)"), I am concerned a possible discrepancy between the sentence in L160-162 ("Further sensitivity tests were performed ... did not change"). I am wondering that if the ATL and PAC domains are shifted westward or eastward for all the seasons, how will the results change? I'd like to know how are sensitive the eddy feedback effects to the relative positions to the storm tracks in each season and each region (ATL and PAC).

Thank you for pointing out this inconsistency in the text, and we agree that this idea is something that warrants further investigation. The sensitivity of our results to the position of the ATL and PAC

domains have been explored further, by shifting both domains to the east and the west by 30 degrees, and producing the same plots as Figs. 7, 8, 9 in the main text. These will be discussed in more detail below. Table AR1 shows the coordinates of the shifted domains. It is worth highlighting that by shifting the domains in this way, in most cases different blocking centres would be analysed (e.g. Urals in the ATL east shift domain), and therefore we would be able to conclude less about Pacific and Euro-Atlantic blocking, which is the main focus of the study.

**Table AR1.** *Details of the ATL and PAC regions used when testing the sensitivity of our results to the positioning of the domains.*

| Domain | Coordinates | Notes |
|---|---|---|
| ATL (west shift) | 60°W-0°E, 45-75°N | Positioned almost exclusively over the North Atlantic; Greenland blocking would be wholly within this domain. |
| ATL | 30°W-30°E, 45-75°N | As used in the paper. Focused on the broad DJF climatological blocking maximum over NE Atlantic and NW Europe (Fig. 5a), but also covers regions of large year-round blocking frequency. |
| ATL (east shift) | 0-60°E, 45-75°N | The focus for this domain would be a mixture of Scandinavian and Ural blocking. Ural blocking is not included in the ATL domain in the paper, but as shown in Fig. 5, there is another climatological blocking maximum here, year-round. |
| PAC (west shift) | 160°E-80°W, 40-70°N | The JJA blocking maximum in the North Pacific is covered better in this domain. |
| PAC | 170-110°W, 40-70°N | As used in the paper. Centred on the DJF North Pacific blocking maximum (Fig. 5a), but also positioned to consider blocks with the potential to cause large societal impacts by overlapping with some of continental North America (see also response to Reviewer #1 above). |
| PAC (east shift) | 140-80°W, 40-70°N | More blocks that impact North America are covered, but coincide with very low block frequencies in JJA. |

Relationship between Number of AC Eddies and Block Persistence, with Shifted Domains

*ATL (west shift), Figs. AR2a, c, e:* Very similar correlations between the number of AC eddies (N), block area (A), and block persistence (P) are found in DJF when the ATL domain is shifted to the west compared to the ATL domain in the paper (Fig. 7a). In JJA however, corr(N,P) in the west-shifted ATL domain is half of that in the ATL domain, and corr(N,A) is not statistically significant. This perhaps suggests that for summer oceanic Atlantic/Greenland blocking, the number of AC eddies is not very important for determining block persistence. This can also be derived from Fig. AR2e, where the mean number of AC eddies for JJA blocks is broadly the same for all persistences.

*PAC (west shift), Figs. AR2b, d, f:* Corr(N,P) and corr(P,A) are almost identical to those in the PAC domain in the paper in both DJF and JJA (Fig. 7b, d). Corr(N,A) is larger in the west-shifted PAC domain. Mean AC eddies per block are also broadly similar to those in the centred-PAC.

*ATL (east shift), Figs. AR3a, c, e:* Again, the relationships between N, P, and A in the east-shifted ATL domain are strong. In JJA, the relationships are smaller but again fairly similar to those in the ATL domain, and unlike when shifted west, the mean number of AC eddies in blocks does increase more noticeably as P increases.

[Figure]

[Figure]

[Figure]

[Figure]

**Figure AR2.** *As in Fig. 7 in the main text, but for the west-shifted domains. Statistically insignificant correlations are depicted by italic font.*

**Figure AR3.** *As in Fig. AR2, but for the east-shifted domains. Statistically insignificant correlations are depicted by italic font.*

*PAC (east shift), Figs. AR3b, d, f:* Corr(N,P) in DJF is slightly lower in the east-shifted PAC domain, compared to the one used in the paper, but mean number of AC eddies per block still increases as block persistence increases, just to a lesser extent. Correlations in JJA are all smaller than what is found in the actual PAC region (and corr(N,A) is statistically insignificant) – this is due to the low climatological blocking frequency over continental North America and thus low sample size.

Relationship between AC Eddy Strength and Block Persistence, with Shifted Domains

*DJF ATL (west shift) and PAC (west shift), Fig. AR4:* When the domains are shifted to the west, broadly the same relationships present themselves as the conclusions in the text. The intensification of the AC eddies once inside the blocks is slightly larger in both west-shifted domains than the original (especially in the west-shifted PAC), however AC eddies contributing to the longest blocks still intensify more than those contributing to shorter blocks. AC eddy speed before/during/after blocking in both west-shifted domains is also qualitatively similar to that in the original PAC and ATL domains.

*JJA ATL (west shift) and PAC (west shift), Fig. AR5:* Results for the west-shifted ATL domain for AC eddy strength and speed are almost identical to those presented in the main text for the original ATL domain. The west-shifted PAC domain now also displays results that are consistent with the ATL domain – and this is already mentioned in the Discussion section of the main text, so is not discussed further here.

*DJF ATL (east shift) and PAC (east shift), Fig. AR6:* Findings from the east-shifted ATL domain are qualitatively similar to those in the original ATL domain. However, when the PAC domain is shifted to the east, there is no statistically significant different in the strength of the AC eddies that contribute to the longest and shortest blocks. We note however that this east-shifted PAC domain is cutting off

part of the climatological blocking maximum, so certain events may only partially being captured by this domain.

*JJA ATL (east shift) and PAC (east shift), Fig. AR7:* AC eddy speed is still consistent with the results for the original ATL and PAC domains. However, AC eddy intensity for the east-shifted ATL domain now demonstrates similar behaviour than in DJF (i.e. stronger eddies result in longer blocks), which was not the case in the original ATL domain. This east-shifted domain is however to the east of the Atlantic JJA blocking frequency maximum, so it is again possible that different block dynamics are being investigated when shifting the domain to the east, rather than in the original ATL domain. AC eddies contributing to east-shifted PAC blocks show no relationship between strength and block persistence, however the low climatological blocking frequency here means caution must be given while interpreting this result.

Summary of Shifting the Domains

We appreciate this was a useful exercise to undertake, and again thank the reviewer for suggesting we investigate this. Broadly speaking, we conclude that our results are fairly insensitive to the

[Figure]

**Figure AR4.** *As in Fig. 8 in the main text, but for the west-shifted domains in DJF.*

**Figure AR5.** *As in Fig. 9 in the main text, but for the west-shifted domains in JJA.*

[Figure]

**Figure AR6.** *As in Fig. 8 in the main text, but for the east-shifted domains in DJF.*

**Figure AR7.** *As in Fig. 9 in the main text, but for the east-shifted domains in JJA.*

positioning of our ATL and PAC domains, provided that climatological blocking frequencies are relatively high within them. Therefore, if we were purely looking at the dynamics (without considering the potential impacts), moving the PAC domain in JJA in particular to align more with the seasonal blocking maximum would yield similar results. Furthermore, as previously mentioned in the text, and in response to Reviewer #1, we positioned the original ATL and PAC domains to give us an insight into the block events that have the potential to cause considerable impacts to society (the west-shifted PAC domain is almost entirely in the ocean, so these blocks would not cause as many impacts as those in the original). Thus, we will keep the ATL and PAC domains as they are in the original text. We will however clarify the contradiction relating to the domains in the text saying that "Further sensitivity tests were performed ... did not change". We will remove this sentence, and in the Discussion section we add a further comment about how results being potentially clearer if domains are located to align more with climatological maxima: "Clearer relationships between AC eddy strength and block persistence could be produced when the PAC (and to a lesser extent, the ATL domain) are aligned more with the seasonal climatological block frequency maxima, however the results presented here are still important since the blocks analysed have the potential to cause more impacts than those e.g. over the Pacific Ocean. The methodology presented here can be applied robustly anywhere, provided that climatological block frequency is relatively high."

Other minor comments:
- L212-213 and L280: I am wondering that what do you specify the "upstream forcing" here? I feel usually the upstream forcing corresponds to eddy-feedback or incoming Rossby-wave (lower frequency) one but they tend to reinforce blocking rather than dissipate and advect downstream?

We appreciate there may be confusion in using this phrase. We have adapted the sentences in the text to remove the phrase "upstream forcing" and simply state that larger, stronger blocks take longer to advect downstream or dissipate.

- The reviewer prefers to avoid using the term "observe" or "observation" except for actual observation, for example, launching a radiosonde, measuring temperature, etc. I would like to ask please consider rewording those terms, if possible.

Thank you for this suggestion; we have changed "observed" to a more appropriate word where necessary (the "observed" in the very first sentence of the Introduction has been kept since it concerns weather conditions at the surface).

More recently, I have read a manuscript "Transient anticyclonic eddies and their relationship to atmospheric blocking persistence". I found that this manuscript is interesting, but this study is a phenomenological one, which cannot identify the causal relationship between transient anticyclonic eddies and blocking persistence. Because this study is in my research domain and the relation between blocking and transient synoptic eddies has been examined in my group before many years, below I give some comments on this manuscript to help the authors understand how transient eddies reinforce and maintain blocking and how intensified blocking deforms transient eddies.

We thank Dehai Luo for taking interest in our work and taking the time to leave thorough suggestions for further understanding of block-eddy interactions, and we apologise for the late response. Responses to the numbered points are addressed below:

1) In the daily geopotential height field of a blocking flow, the synoptic-scale anticyclonic (cyclonic) eddies are often seen to be intensified and shifted northward (southward) during the blocking growth and maintenance episodes (Berggren et al. 1949). Such an eddy deformation is also referred to as eddy straining or cyclonic wave breaking (CWB). Thus, many investigators inferred that eddy straining or CWB leads to blocking onset. Based on this, they also concluded that persistent eddy straining or persistent CWB leads to persistent blocking. In fact, the relationship between transient eddies and blocking is a chicken-egg problem. Thus, one cannot examine the causal relationship between the blocking persistence and transient anticyclonic eddies only using the identification method. However, this issue can be solved by a nonlinear theoretical model that considers a blocking as a nonlinear initial value issue of the blocking interacting with synoptic-scale eddies. The relationship between blocking and transient synoptic-scale eddies and what determines the persistence of atmospheric blocking has been widely examined in a nonlinear multi-scale interaction (NMI) model and has been clearly clarified (Luo et al. 2005, 2014, 2019).

Thank you for pointing us to these references, as they provide a different perspective on the relationship between block persistence and anticyclonic (AC) synoptic eddies. The work of Luo et al. (2005, 2014, 2019) considers this relation in an idealised model world, whereas our paper here concerns a climatological study using real-world data. While we do not necessarily test for exactly the same mechanisms that are highlighted in Luo's work, we believe that our climatological study is complementary to Luo's findings. The climatological effectiveness of the eddy-block matching (EBM) mechanism, in particular, is something we cannot quite ascertain in our study since we are not looking at the background vorticity field. However, we appreciate that this could potentially be an important process and as such, we have now addressed this in some introductory remarks in the text ("…This concept is expanded further in Luo et al. (2005, 2014, 2019) using the Eddy-Block Matching (EBM) mechanism theory. In the EBM mechanism, the two-way relationship between blocks and synoptic-scale eddies is explained via eddy vorticity forcing (EVF). If the background EVF is favourable for block amplification, the block feeds back onto the eddies to strengthen them and the background EVF, which further amplifies the block, and so on").

2) The mutual relationship between the blocking and synoptic-scale eddies has been examined in Luo (2005). It is found that the blocking and synoptic-scale eddies are dependent each other. Because of the feedback of the intensified blocking on the pre-existing synoptic-scale eddies prior to entering the blocking region, the pre-existing synoptic-scale eddies may slow down and

undergo a north-south straining, which are dominated by deformed eddies with meridional tripoles.

We understand that the AC eddies and blocks have a symbiotic relationship, and we believe the text already discusses this. We have found that the AC eddies act to strengthen and enlarge the block (Fig. 4 in the updated manuscript), while the blocks act to attract the eddies and move them in a northward direction (Fig. 8e-f, 9e-f). The northward movement of the eddies (in addition to being due to the attraction and ridge-building as already mentioned in the text) could also represent the northward component of the meridional eddy straining. The strengthening of the AC eddies that is observed, particularly in the PAC domain, in the ~3 days before entering the block (Fig. 8b, 9b) could be a sign of the block acting the strengthen the upstream eddies as well, agreeing with the theory from Luo (2005). A comment addressing this possibility has been included in the text in Sect. 5.2: "In the PAC region, the intensification is stronger for DJF eddies (nearly 100 m) and begins at around day -3. It is possible that this could be a sign that the block is acting to strengthen the upstream AC eddies, consistent with the EBM mechanism (Luo, 2005). However, it could also potentially because the PAC region is slightly to the east of the North Pacific climatological blocking maximum…"

3) Atmospheric blocking persistence has been first investigated by Yeh (1949), who found that atmospheric blocking tends to be long-lived in high latitudes. Luo et al. (2019) found from the NMI model that when the north-south gradient (PVy) of background potential vorticity is smaller, atmospheric blocking tends to be more persistent. Note that PVy is a modified $\beta_\circ$ When the background westerly wind or meridional temperature gradient is weaker or the latitude is higher, PVy is smaller. In this case, the blocking system has weaker energy dispersion and stronger nonlinearity so that the blocking can be more persistent. Thus, the persistence of atmospheric blocking is mainly determined by the background condition (i.e., the magnitude of PVy), rather than synoptic-scale eddies, even though synoptic-scale eddies have different phase speeds under different background conditions. As also noted by Luo et al (2019), the eddy forcing induced by pre-existing synoptic-scale eddies prior to entering the initial blocking is more persistent as PVy is smaller. In this case, atmospheric blocking can be more persistent due to persistent eddy forcing by pre-existing synoptic eddies prior to entering the initial blocking.

The background $PV_y$, as mentioned in Luo et al. (2019), is typically smaller at higher latitudes, explaining why blocks are more persistent here. However, by fixing our ATL and PAC domains in our study, we are largely taking out the effects of differing background PV in determining block persistence. A weaker background circulation is present in both domains in the summer (which can be inferred from the plots of $\overline{Z_*}$ in Fig. 1), meaning the background $PV_y$ is also weaker in summer. Therefore, one would expect from the NMI model that summer blocks are more persistent than winter ones (when $PV_y$ in the midlatitudes is much stronger). However, we find that blocks are actually least persistent in summer, and most persistent in winter, which contradicts the theory from the NMI model. Therefore, we do not think that the magnitude of the background circulation is an important factor in driving block persistence in our study, and instead conclude that, to first order, block persistence is related in many cases (but not all) by the number of incoming AC eddies (Fig. 7).

4) The interaction between transient eddies and blocking satisfies the symbiotic relation noted by Cai and Mak (1990). The onset and intensification of atmospheric blocking not only depends on the spatial structure of pre-existing synoptic-scale eddies prior to entering the initial blocking, but also the deformation of pre-existing synoptic-scale eddies depends on the intensification of atmospheric blocking. Thus, the blocking and synoptic-scale eddies are coupled together and

dependent each other. For an given initial blocking, the initial blocking can be amplified into a typical blocking if pre-existing synoptic-scale eddies ($\psi_1'$) prior to entering the initial blocking satisfy $\frac{\partial q}{\partial t} \cong -\nabla \cdot (\mathbf{v}_1' q_1')_P$ (Luo et al. 2014), where $q$ is the PV anomaly of the initial blocking, $\mathbf{v}_1' = (-\frac{\partial \psi_1'}{\partial y}, \frac{\partial \psi_1'}{\partial x})$ and $q_1' = \nabla^2 \psi_1'$. In other words, when the eddy forcing $-\nabla \cdot (\mathbf{v}_1' q_1')_P$ has the same spatial structure as the PV anomaly $q$ of the initial blocking, a typical blocking can form from this initial blocking under the eddy forcing. When the initial blocking is intensified (Fig. 4a of Luo et al. 2014), the feedback of intensified blocking can cause the deformation of preexisting synoptic-scale eddies. In this case, deformed eddies ($\psi_2'$) are produced. The daily synoptic-scale eddy field during the blocking episode can be represented by $\psi' = \psi_1' + \psi_2'$. Because $\psi_2'$ includes the amplitude of the intensified blocking, the synoptic-scale eddies in the daily synoptic-scale eddy field ($\psi' = \psi_1' + \psi_2'$) are inevitably intensified, split into two branches and slowed down with the growth of blocking (Fig. 4b of Luo et al. 2014). This case corresponds to eddy straining. In the daily total field (the sum of mean flow, blocking part and $\psi' = \psi_1' + \psi_2'$) of the blocking flow, anticyclonic (cyclonic) eddies are intensified and shifted northward with the blocking growth (Fig. A2), which corresponds to CWB (Fig. 4c of Luo et al. 2014). When the blocking is more persistent (Luo et al. 2019), more transient anticyclonic eddies are seen due to the persistent feedback of blocking because there is a symbiotic relationship between the blocking and anticyclonic eddies (Fig. 4c of Luo et al. 2014). This does not imply that persistent blocking is produced by more anticyclonic eddies.

We thank you for pointing out that there is a positive feedback between the blocks and eddies. After the first round of reviewer comments, we explained this in the Discussion section of the updated manuscript in terms of the Selective Absorption Mechanism (SAM; Yamazaki and Itoh, 2013), though appreciate that a similar argument could also be made through the EBM mechanism. We have added the EBM mechanism suggestion to the discussion of the PAC results, as highlighted above.

5) The authors concluded that blocks can be maintained through repeated absorption of anticyclonic eddies. In fact, blocking events do not always occur, but synoptic scale anticyclonic eddies can often be seen. Why some anticyclonic eddies can be absorbed into the blocking, but others cannot. The authors should answer under what condition the anticyclonic eddies can be absorbed into the blocking to maintain it. This problem is easily explained in terms of $\frac{\partial q}{\partial t} \cong -\nabla \cdot (\mathbf{v}_1' q_1')_P$ because only some of synoptic-scale eddies can meet this condition. When the preexisting synoptic-scale eddies ($\psi_1'$) drive the onset and intensification of blocking ($q$), the feedback of blocking can cause the deformation of preexisting synoptic-scale eddies to result in the repeated absorption of anticyclonic (cyclonic) eddies by the blocking (Fig. A2 or Fig. 4c of Luo et al. 2014). When the blocking has longer lifetime, there are inevitably more anticyclonic (cyclonic) eddies within the blocking regions. Thus, persistent blocking can often occur together with more anticyclonic (cyclonic) eddies. But, this cannot lead us to conclude that more anticyclonic (cyclonic) eddies lead to persistent blocking.

We do not consider the AC eddies that do not interact with blocks, as one of our main goals was to examine the climatological relationship between AC eddy-block interactions and how these interactions determine block persistence. Upstream AC eddies that do not interact with an already-present block were therefore out of scope for this study. As our study only selects those AC eddies that do interact with blocks, while neglecting those that don't, we can say with confidence how these AC eddies may lead to more persistent blocking in support of our hypothesis (see again Fig. 7).

I suggest that the authors should read the following references to improve the understanding of how the blocking and synoptic-scale eddies interact and what leads to the persistence of atmospheric blocking.

References:

Berggren, R., Bolin, B. and C.-G., Rossby, 1949: An aerological study of zonal motion, its perturbations and break-down. Tellus, 1, 14–37.

Cai, M and M. Mak, 1990: Symbiotic relation between planetary and synoptic-Scale waves. J. Atmos. Sci., 47, 2953–2968

Luo, D., 2005: A barotropic envelope Rossby soliton model for block-eddy interaction. Part III: Wavenumber conservation theorems for isolated blocks and deformed eddies, J. Atmos. Sci., 62, 3839-3859

Luo, D., J. Cha, L. Zhong, and A. Dai, 2014: A nonlinear multiscale interaction model for atmospheric blocking: The eddy-blocking matching mechanism. Quart. J. Roy. Meteor. Soc., 140, 1785–1808, doi:10.1002/qj.2337.

Luo, D., W. Zhang, L. Zhong and A. Dai, 2019: A nonlinear theory of atmospheric blocking: A potential vorticity gradient view. J. Atmos. Sci., 76, 2399-2427. Yeh, T. C., 1949: On energy dispersion in the atmosphere. J. Meteor., 6, 1-16.

Thank you for pointing us to these references for further background reading. We have added the references of Luo to our introduction. We also reference the Yamazaki and Itoh (2013) paper, mentioned in our response to point (4).

Yamazaki, A. and Itoh, H., 2013. Vortex–vortex interactions for the maintenance of blocking. Part I: The selective absorption mechanism and a case study. Journal of the Atmospheric Sciences, 70(3), pp.725-742.